# FORWARD GRADIENT TRAINING OF SPIKING NEURAL NETWORKS

## ABSTRACT

Neuromorphic computing with spiking neural networks (SNNs) is promising for energy-efficient applications. However, the supervised learning of SNNs is challenging considering biological plausibility and neuromorphic hardware compatibility. Most existing successful methods rely on backpropagation (BP) through time and across layers for temporal and spatial credit assignments, which is hard to realize. While some online training methods tackle temporal credit assignment by eligibility traces, it remains an important problem for error signal propagation with proper spatial credit assignment. In this work, we propose a new method, forward gradient training (FGT), for spiking neural networks. FGT only leverages unidirectional forward propagation across layers and direct feedback signals from the top layer to calculate gradients for spatial credit assignment, and we improve the large variance of vanilla forward gradients by momentum feedback connections. FGT avoids layer-by-layer forward-backward calculation of BP with symmetric weights and separate phases, and has more theoretical guarantee and better performance compared with random feedback methods. When combined with online training methods, FGT enables forward and online training. This paves solid paths to on-chip SNN training. Extensive experiments demonstrate the effectiveness and robustness of FGT with similar performance as BP under both fully connected and convolutional networks on static and neuromorphic datasets.

## 1 INTRODUCTION

Brain-inspired neuromorphic computing with biologically plausible spiking neural networks (SNNs) has gained increasing attention in recent years (Roy et al., 2019). SNNs imitate biological spiking neurons to transmit spike trains for event-based computation, which enables efficient and parallel in-memory computation on neuromorphic hardware with low energy consumption (Akopyan et al., 2015; Davies et al., 2018; Pei et al., 2019; Woźniak et al., 2020; Rao et al., 2022).

However, the supervised training of neuromorphic computing systems is a challenging problem, especially considering biological plausibility and hardware compatibility. While the problem of discrete spike-generation procedures can be solved by surrogate gradient or other methods as shown in recent works (Shrestha & Orchard, 2018; Wu et al., 2018; Neftci et al., 2019), these methods rely on backpropagation (BP) through time and across layers, which is biologically implausible and unfriendly for neuromorphic hardware.

**Limitation of BP** Specifically, the problem is how to solve temporal and spatial credit assignments. BP through time deals with temporal credit assignment by retaining the whole computational graph of previous time steps and propagating errors along the reverse time, which is inconsistent with the online property of spiking neurons (Bellec et al., 2020; Xiao et al., 2022). BP across layers tackles spatial credit assignment by backward layer-by-layer error propagation. Considering the unidirectional synapses between neurons, this requires reciprocal connections with symmetric weights, as well as separate phases for signal propagation and storage of (surrogate) derivatives during forward propagation for calculation of backward propagation. These are often regarded as biologically implausible (Nøkland, 2016) and pose great challenges for on-chip training of SNNs.

Many previous works attempt to solve the problem with biologically more plausible training methods. For temporal credit assignment, online training methods based on eligibility traces have been proposed (Bellec et al., 2020; Xiao et al., 2022). These methods track traces of activations to decouple the temporal dependency and enable forward-in-time learning given error signals. Considering

spatial credit assignment for error signals of multi-layer networks, however, the problem is much tougher to tackle. Some online training methods still rely on BP across layers (Bohnstingl et al., 2022; Xiao et al., 2022). Existing works considering global supervision without BP either only relieve the weight symmetric problem with target propagation (Lee et al., 2015), random/learned feedback (Lillicrap et al., 2016; Akrout et al., 2019), or sign symmetric (Xiao et al., 2018), but still require layer-by-layer backward propagation with reciprocal connections, separate phases, and storage of derivatives, or directly propagate errors from the top layer to hidden layers by random feedback weights, e.g., direct feedback alignment (DFA) (Nøkland, 2016), with limited guarantee and poorer performance than BP. Some recent works propose forward gradients (Silver et al., 2022; Baydin et al., 2022), but they suffer from large variances and poor performance. Some other works turn to local learning with local readout layers for supervision (Kaiser et al., 2020) or local forward-forward contrastive self-supervised learning (Hinton, 2022; Ororbia, 2023). As the basic component of modern artificial intelligence, global learning is crucial to achieving promising results, and it can also be combined with local learning. So it remains an important question to investigate better spatial credit assignment methods for global learning.

**Contribution**  In this work, we propose a new method, forward gradient training (FGT), for spiking neural networks, which only requires unidirectional forward propagation across layers and feedback signals directly from the top layer for spatial credit assignment. Specifically, inspired by the recent work of forward gradient (Baydin et al., 2022), we first propose forward surrogate gradients to estimate backpropagated surrogate gradients in an unbiased approach with perturbation vectors. Then, to tackle the large variance of vanilla forward gradient, we propose forward gradient with momentum feedback connections, which maintains feedback weights based on forward propagated perturbation vectors for direct error propagation from the top layer to hidden middle ones. Combining FGT with online training methods for spatiotemporal credit assignments, we can obtain forward and online training methods for SNNs, which pave solid paths for on-chip SNN training. Compared with previous biologically plausible training methods, our method does not need layer-by-layer backward propagation with separate phases or unguaranteed random feedback, and significantly outperforms random feedback methods. Our contributions include:

1. We propose forward surrogate gradients for SNNs, which only requires unidirectional forward propagation across layers and are unbiased estimator of backpropagated gradients.

2. We further propose forward gradient with momentum feedback connections to reduce the variance and stabilize training, which maintains feedback weights directly from the top layer to hidden layers. The method provides a more biologically plausible spatial credit assignment method than BP with more guarantee than random feedback. Combined with online training, it can pave paths for on-chip SNN learning.

3. We conduct extensive experiments on MNIST, N-MNIST, DVS-CIFAR10, DVS-Gesture, CIFAR-10, and CIFAR-100 with both fully connected and convolutional networks, which demonstrate the effectiveness of our method to reach a similar performance as BP. Experiments also show the effectiveness of variants considering different signal propagation methods, local learning, and stochastic neuron models, revealing the robustness and generalization ability of our method.

## 2  RELATED WORK

**SNN Training Methods**  For supervised training of SNNs, backpropagation through time is the commonly used framework and the non-differentiable problem of the spiking function is solved by applying surrogate derivatives (Shrestha & Orchard, 2018; Wu et al., 2018; 2019; Neftci et al., 2019; Li et al., 2021; Deng et al., 2022) or computing gradients with respect to spiking times (Zhang & Li, 2020; Kim et al., 2020; Zhu et al., 2022). These methods rely on BP through time and across layers. Another direction is to build connections between SNNs and equivalent closed-form mappings (or implicit equilibriums) similar to artificial neural networks (ANNs) with specific encodings of spike trains (e.g., firing rates or the first time to spike), and convert ANNs to SNNs (Rueckauer et al., 2017; Sengupta et al., 2019; Deng & Gu, 2021; Stöckl & Maass, 2021; Meng et al., 2022b) or train SNNs by gradients calculated from the mappings (Lee et al., 2016; Zhou et al., 2021; Wu et al., 2021; Meng et al., 2022a) or equilibriums (O'Connor et al., 2019; Xiao et al., 2021; Martin et al., 2021; Xiao et al., 2023). These methods also rely on BP across layers for multi-layer networks or are limited to single-layer networks with symmetric recurrent connections. Considering biologically

more plausible / hardware-friendly methods, Zenke & Ganguli (2018); Bellec et al. (2020); Bohnstingl et al. (2022); Xiao et al. (2022); Yin et al. (2021) propose online training methods rather than BP through time by tracking eligibility traces or adding regularization for forward-in-time learning, but they require BP across layers or leveraging random feedback with limited guarantee. Neftci et al. (2017); Lee et al. (2020) apply DFA to SNNs with random feedback as well. Kaiser et al. (2020) proposes online local learning of SNNs by ignoring temporal dependencies, and Yang et al. (2022) proposes local tandem learning with ANN teachers. Different from them, we propose a new method for spatial credit assignment of global learning without BP while maintaining similar performance.

**Neural Network Training without Backpropagation** Many previous works explore methods other than BP across layers for effective global training of neural networks. Target propagation (Lee et al., 2015) propagates targets instead of errors to avoid the weight symmetric problem. Feedback alignment (Lillicrap et al., 2016) replaces backward weights between successive layers with random matrices. Akrout et al. (2019) improves it by learning backward weights to be symmetric with forward weights. Sign symmetric (Liao et al., 2016; Xiao et al., 2018) shares the sign between backward and forward weights. These methods still require layer-by-layer propagation as BP. Direct feedback alignment (Nøkland, 2016; Launay et al., 2020) pushes forward feedback alignment to directly propagate errors from the top layer to hidden layers with random weights. However, methods with random feedback are with limited guarantee and usually perform worse than BP, especially for convolutional networks. Webster et al. (2020) borrows the idea from Akrout et al. (2019) to learn feedback weights in DFA, and Bacho & Chu (2022) proposes to learn them with directional derivatives. These works are for ANNs rather than SNNs and are without much theoretical justification. Silver et al. (2022) and Baydin et al. (2022) propose unbiased random directional gradients by forward computation, which inspire this work, but they cannot achieve competitive performance due to the large variance. Ren et al. (2023) improves forward gradients with various architecture designs and local losses, but still has a performance gap with BP and is orthogonal to our momentum feedback connections. There are also other methods such as equilibrium propagation (Scellier & Bengio, 2017) to train neural networks with energy functions, but they are limited to symmetric connections and usually perform worse. Some methods use lifted proximal formulation to train neural networks without explicit BP (Li et al., 2020). Other works turn to local learning rather than global supervision. Kaiser et al. (2020) leverages local readout layers for supervision. Hinton (2022) proposes forward-forward contrastive self-supervised learning for layer-wise gradual learning. Different from them, our work focuses on global learning and can also be combined with local learning.

## 3 PRELIMINARIES

### 3.1 SPIKING NEURAL NETWORKS

SNNs are composed of brain-inspired spiking neurons that transmit information by spike trains. Each neuron maintains a membrane potential $u$ to integrate input spike trains, and generates a spike once $u$ exceeds a threshold, after which $u$ is reset to the resting potential. We consider the commonly used leaky integrate and fire (LIF) model with the dynamics of the membrane potential as:

$$\tau_m \frac{du}{dt} = -(u - u_{rest}) + R \cdot I(t), \quad u < V_{th}, \tag{1}$$

where $I$ is the input current, $V_{th}$ is the threshold, and $R$ and $\tau_m$ are resistance and time constant, respectively. When $u$ reaches $V_{th}$ at time $t^f$, a spike is generated and $u$ is reset as zero. The output spike train is defined with the Dirac delta function: $s(t) = \sum_{t^f} \delta(t - t^f)$.

A spiking neural network consists of connected spiking neurons with weights. We consider the simple current model $I_i(t) = \sum_j w_{ij} s_j(t) + b_i$ (the subscript $i$ represents the $i$-th neuron), where $w_{ij}$ is the weight from neuron $j$ to neuron $i$, and $b_i$ is a bias. The discrete computational form is:

$$\begin{cases} u_i[t+1] = \lambda(u_i[t] - V_{th} s_i[t]) + \sum_j w_{ij} s_j[t] + b_i, \\ s_i[t+1] = H(u_i[t+1] - V_{th}), \end{cases} \tag{2}$$

where $H(x)$ is the Heaviside step function, $s_i[t]$ is the spike train of neuron $i$ at discrete time step $t$, $\lambda < 1$ is a leaky term (typically taken as $1 - \frac{1}{\tau_m}$). The constant $R$, $\tau_m$, and time step size are absorbed into the weights and bias. For the discrete time steps in multi-layer networks, we use $\mathbf{s}^{l+1}[t]$ to denote the $(l+1)$-th layer's response after receiving the $l$-th layer's signals $\mathbf{s}^l[t]$, i.e., the expression is $\mathbf{u}^{l+1}[t+1] = \lambda(\mathbf{u}^{l+1}[t] - V_{th}\mathbf{s}^{l+1}[t]) + \mathbf{W}^l\mathbf{s}^l[t+1] + \mathbf{b}^l$.

Most spiking neuron models consider the deterministic condition above, while biological stochastic spiking can be modeled as well. In the stochastic setting, a neuron generates spikes following a Bernoulli distribution, whose probability is the c.d.f. of a distribution w.r.t $u[t] - V_{th}$, i.e., $f(u_i[t+1] - V_{th})$ that allows higher probability for a spike with larger $u[t] - V_{th}$. This can be reparameterized as the spiking function $s_i[t+1] = H(u_i[t+1] - V_{th} - z_i)$, where $z_i$ is a random variable following the distribution specified by $f$. Under the stochastic setting, surrogate gradients of SNNs can be introduced in a systematic approach (Shekhovtsov & Yanush, 2021; Shekhovtsov et al., 2020). Stochastic neuron models may better tackle the hardware noise, e.g., thermal noise or neuron silencing (Büchel et al., 2021), and can be more robust (Ma & Tang, 2022). We consider the deterministic setting by default and will show that FGT is also applicable to the stochastic setting.

## 3.2 Surrogate Gradient of SNNs

Due to discrete spikes, training SNNs is a hard problem. A popular approach is the surrogate gradient (SG) method combined with backpropagation through time (BPTT) or its variants. BPTT unfolds the iterative update equation in Eq. (2) and backpropagates along the computational graph (Fig. 1(a)). The non-differentiable terms $\frac{\partial \mathbf{s}^l[t]}{\partial \mathbf{u}^l[t]}$ will be replaced by surrogate derivatives of a smooth function, e.g., that of the sigmoid function: $\frac{\partial s}{\partial u} = \frac{1}{a_1} \frac{e^{(V_{th}-u)/a_1}}{(1+e^{(V_{th}-u)/a_1})^2}$ ($a_1$ is hyperparameter).

Some works propose temporally online training methods by tracking eligibility traces of neurons to overcome the drawback of BPTT. Take the recent OTTT method (Xiao et al., 2022) as an example, gradients are online calculated at each time by the tracked presynaptic trace $\hat{\mathbf{a}}^l[t] = \sum_{\tau \leq t} \lambda^{t-\tau} \mathbf{s}^l[\tau]$ and instantaneous gradient $\mathbf{g}_{\mathbf{u}^{l+1}}[t] = \left( \frac{\partial \mathcal{L}[t]}{\partial \mathbf{s}^N[t]} \prod_{i=0}^{N-l-2} \frac{\partial \mathbf{s}^{L-i}[t]}{\partial \mathbf{s}^{L-i-1}[t]} \frac{\partial \mathbf{s}^{l+1}[t]}{\partial \mathbf{u}^{l+1}[t]} \right)^{\top}$ as $\nabla_{\mathbf{W}^l} \mathcal{L}[t] = \mathbf{g}_{\mathbf{u}^{l+1}}[t] \hat{\mathbf{a}}^l[t]^{\top}$, where $\frac{\partial \mathbf{s}^l[t]}{\partial \mathbf{u}^l[t]}$ is also replaced by surrogate derivatives. They mainly solve the problem of temporal BP (Fig. 1(a,b)), but still require spatial BP with reciprocal connections between successive layers for spatial credit assignment of multi-layer networks (Fig. 1(c)).

**Theoretical Grounding of Surrogate Gradient** SG for the deterministic spiking model is usually considered empirical. Xiao et al. (2022) attempted to connect surrogate derivatives with derivatives of well-defined transformations under their OTTT method, with a certain surrogate function and convergent inputs. Considering stochastic neuron models, however, SG can be viewed as a systematic gradient estimation (Shekhovtsov & Yanush, 2021; Shekhovtsov et al., 2020). Specifically, for stochastic models with the objective function of its expectation, gradients are actually well-defined and the form of SG can be derived with techniques including derandomization and linear approximation, in which surrogate derivatives correspond to the probabilistic density function of the neuron model. The deterministic condition may be viewed as a special case. More introductions are in Appendix A. Overall, we consider SG as a reliable target for our forward gradient to estimate.

## 3.3 Forward Mode Automatic Differentiation

Traditional backpropagation is based on the reverse mode automatic differentiation (AD), i.e., after the forward calculation $\mathbf{x} \to \mathbf{f}(\mathbf{x})$, the error $\mathbf{e}$ is propagated in the backward pass $\mathbf{J_f}(\mathbf{x})^{\top}\mathbf{e} \leftarrow \mathbf{e}$, where $\mathbf{J_f}(\mathbf{x})$ is the Jacobian of $\mathbf{f}$ evaluated at $\mathbf{x}$. For example, for the linear transformation $\mathbf{f}(\mathbf{x}) = \mathbf{Wx}$, the error is backpropagated as $\mathbf{W}^{\top}\mathbf{e}$. This leads to the forward-backward calculation of BP with symmetric weights. The forward mode AD, on the other hand, only requires forward calculation. Given a perturbation vector $\mathbf{v}$, during the forward calculation $\mathbf{x} \to \mathbf{f}(\mathbf{x})$, forward AD simultaneously calculate the Jacobian-vector product $\mathbf{v} \to \mathbf{J_f}(\mathbf{x})\mathbf{v}$. For the case of $f : \mathbb{R}^n \to \mathbb{R}$, the product is the directional derivative $\langle \nabla f(\mathbf{x}), \mathbf{v} \rangle$, and the forward gradient (Baydin et al., 2022) can be defined as $\mathbf{g}(\mathbf{x}) = \langle \nabla f(\mathbf{x}), \mathbf{v} \rangle \mathbf{v}$, which corresponds to the projected gradient in the direction of $\mathbf{v}$. The forward AD avoids the forward-backward calculation and the weight symmetry problem, since the calculation only involves unidirectional calculation: for example, for the linear transformation $\mathbf{f}(\mathbf{x}) = \mathbf{Wx}$, the forward AD similarly calculates $\mathbf{Wv}$.

## 4 Forward Gradient Training of SNNs

In this section, we introduce the proposed forward gradient training of SNNs. We first introduce forward surrogate gradients in Section 4.1. Then in Section 4.2, we introduce forward gradient with momentum feedback connections to tackle the problem of vanilla forward gradient training. In section 4.3, we present combination with online training methods for forward and online training of SNNs. Finally, discussions and implemented details are presented in Section 4.4.

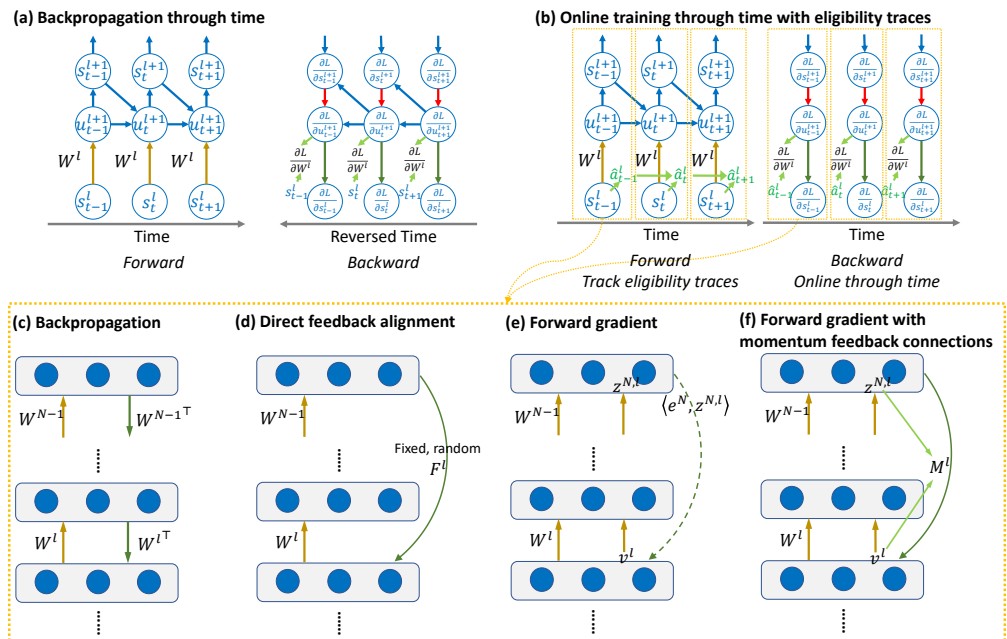

Figure 1: Illustration of SNN training methods. (a-b) Temporal credit assignment. (a) BPTT stores and propagates along the computational graph of previous time, while (b) temporally online methods leverage eligibility traces for forward-in-time learning (Bellec et al., 2020; Xiao et al., 2022). (c-f) Spatial credit assignment. (c) BP backward propagates errors layer-by-layer with symmetric weights. (d) DFA (Nøkland, 2016) directly propagates signals from the top layer to middle ones, but the connections are fixed random matrices with limited guarantee. (e) Forward gradient forward propagates perturbation vectors simultaneously with the forward propagation of neural networks, and afterward, a scalar signal is passed to the middle layer which is shared by all neurons. (f) Forward gradient with momentum feedback connections further tracks feedback connections based on perturbation vectors and directly propagates errors to neurons with top-down connections.

## 4.1 FORWARD SURROGATE GRADIENT

As introduced in Section 3.3, forward gradients can be calculated based on forward mode AD which avoids the drawbacks of BP. The common definition of forward gradients relies on differentiable functions, which is not the case for SNNs. Therefore, we propose forward surrogate gradients for SNNs. We will build our work on online training methods, so the gradients mainly refer to instantaneous gradients at each time step. In the following, we first consider a single time step at $t$.

For common surrogate gradients with BP across layers, the gradient for the $(l+1)$-th layer's neuron output is calculated by $\widetilde{\nabla_{\mathbf{s}^{l+1}}\mathcal{L}}[t] = \left(\frac{\partial\mathcal{L}[t]}{\partial\mathbf{s}^N[t]}\prod_{i=0}^{N-l-2}\frac{\partial\mathbf{s}^{N-i}[t]}{\partial\mathbf{u}^{N-i}[t]}\frac{\partial\mathbf{u}^{N-i}[t]}{\partial\mathbf{s}^{N-i-1}[t]}\right)^{\top}$ with surrogate derivatives of $\frac{\partial\mathbf{s}^{N-i}[t]}{\partial\mathbf{u}^{N-i}[t]}$. This requires backpropagating errors from the last layer to the $(l+1)$-th layer (Fig. 1(c)), i.e., the errors $\mathbf{e}^i = \frac{\partial\mathcal{L}[t]}{\partial\mathbf{s}^i[t]}$ are recursively calculated by

$$\mathbf{e}^i = \mathbf{W}^{i^{\top}}(\mathbf{e}^{i+1}\odot f(\mathbf{u}^{i+1}[t])), \quad i = N-1,\cdots,l+1, \tag{3}$$

where $f(\mathbf{u}^{i+1}[t])$ is the surrogate derivative based on membrane potential in the forward calculation.

We use forward gradients to estimate $\widetilde{\nabla_{\mathbf{s}^{l+1}}\mathcal{L}}[t]$. Specifically, for the $(l+1)$-th layer's neuron output, we will sample a random vector $\mathbf{v}^{l+1}$ with the same size and forward propagate it to the last layer based on the recursive calculation (with $\mathbf{z}^{l+1,l+1} = \mathbf{v}^{l+1}$):

$$\mathbf{z}^{i+1,l+1} = \mathbf{W}^i\mathbf{z}^{i,l+1}\odot f(\mathbf{u}^{i+1}[t]), \quad i = l+1,\cdots,N-1. \tag{4}$$

At the last layer (e.g., the classification output), with the local gradient $\mathbf{e}^N$ obtained by differentiating the loss function, the directional derivative is calculated by $\left\langle\mathbf{e}^N, \mathbf{z}^{N,l+1}\right\rangle$. This process corresponds to calculating the Jacobian-vector product as in Section 3.3. Then we obtain the forward gradient

$$\mathbf{g}_{\mathbf{s}^{l+1}} = \left\langle\mathbf{e}^N, \mathbf{z}^{N,l+1}\right\rangle\mathbf{v}^{l+1}. \tag{5}$$

Similar to previous works (Baydin et al., 2022), we can show that it is an unbiased estimator.

**Property 1.** *When $\mathbf{v}^{l+1}$ has i.i.d. components with zero mean and unit variance, the forward surrogate gradient $\mathbf{g}_{\mathbf{s}^{l+1}}$ is an unbiased estimator of the surrogate gradient $\widetilde{\nabla}_{\mathbf{s}^{l+1}}\mathcal{L}[t]$.*

The distribution of $\mathbf{v}$ has various choices such as the multivariate standard Gaussian distribution (Baydin et al., 2022). Belouze (2022) shows that the centered Rademacher distribution, i.e., $v$ will take $1$ or $-1$ both with the probability $0.5$, can achieve the minimally deviating forward gradients. So we will consider this distribution for $\mathbf{v}$, which also better fits the spike signals of SNNs.

The calculation of forward gradients is simultaneous and coupled with the forward propagation of neural networks. Then the supervision signal $\left\langle \mathbf{e}^N, \mathbf{z}^{N,l+1} \right\rangle$ at the output layer is passed to the $(l+1)$-th layer and formulates the gradient with the local vector $\mathbf{v}^{l+1}$, corresponding to a top-down global modulation signal which is shared by all neurons in the layer, as shown in Fig. 1(e).

Although this forward gradient is an unbiased estimator, it can have a large variance when the number of neurons is large — due to the curse of dimensionality, there is an exponentially growing number of the possible projection direction $\mathbf{v}$. As will be shown in the experiments, the simple forward gradient method cannot work well for a relatively large amount of neurons, and this cannot be directly solved by sampling a few directions for variance reduction due to the high dimensionality.

### 4.2 Forward Gradient with Momentum Feedback Connections

To tackle the problem of vanilla forward gradient training, we propose to leverage momentum feedback connections so that different samples of $\mathbf{v}$ for different inputs can be leveraged to reduce the large variance of a single direction. Specifically, for one sample of $\mathbf{v}$ in each iteration, the forward gradient is calculated by $\mathbf{g}_{\mathbf{s}^{l+1}} = \left\langle \mathbf{e}^N, \mathbf{z}^{N,l+1} \right\rangle \mathbf{v}^{l+1} = \mathbf{v}^{l+1} \mathbf{z}^{N,l+1^\top} \mathbf{e}^N$, which can also be viewed as propagating top error signals $\mathbf{e}^N$ with a connection weight $\mathbf{v}^{l+1}\mathbf{z}^{N,l+1^\top}$ to the middle layer. We can track the momentum

$$\mathbf{M}^{l+1} = \lambda\mathbf{M}^{l+1} + (1-\lambda)\mathbf{v}^{l+1}\mathbf{z}^{N,l+1^\top} \tag{6}$$

and leverage the momentum term for signal propagation to reduce the variance, i.e., the error signals are propagated by

$$\overline{\mathbf{g}_{\mathbf{s}^{l+1}}} = \mathbf{M}^{l+1}\mathbf{e}^N, \tag{7}$$

as shown in Fig. 1(f). The connection $\mathbf{M}^{l+1}$ between the $(l+1)$-th layer and the output layer is first updated by local vectors $\mathbf{v}^{l+1}$ and $\mathbf{z}^{N,l+1^\top}$ of two layers after the forward propagation, and then directly propagates top error signal to the $(l+1)$-th layer.

Compared with vanilla forward gradient, momentum feedback connections can better utilize different samples of $\mathbf{v}$ to reduce the variance and stabilize training. Also, it enables a more detailed spatial credit assignment as it propagates different feedback signals to each neuron, while the feedback scalar signal in the vanilla forward gradient is shared by all neurons in the layer and the difference between neurons comes from the sampled $\mathbf{v}$ which can be imprecise, as shown in Fig. 1(e, f).

However, the momentum may introduce bias as the Jacobian may vary for different inputs, i.e., for different inputs the final $\mathbf{z}^{N,l+1}$ can be different given the same $\mathbf{v}^{l+1}$. This is not related to linear operations but mainly due to the difference in data-dependent neural activations and surrogate derivatives, which can be a shared problem for methods without layer-by-layer error propagation combined with (surrogate) derivatives of each layer.

Actually, the momentum can be viewed as a practical approximation to the expectation of (surrogate) Jacobian over the dataset, and this is leveraged for error propagation instead of data-dependent Jacobian. Let $\widetilde{\mathbf{J}_{\mathbf{f}}}(\mathbf{x})$ denote the surrogate Jacobian of the forward function with surrogate derivatives evaluated with input $\mathbf{x}$. Using the expectation changes the original surrogate gradient $\mathbb{E}_{\mathbf{x}_i}\widetilde{\mathbf{J}_{\mathbf{f}}}(\mathbf{x}_i)\mathbf{e}(\mathbf{x}_i)$ to $\mathbb{E}_{\mathbf{x}_i}\left(\mathbb{E}_{\mathbf{x}_j}\widetilde{\mathbf{J}_{\mathbf{f}}}(\mathbf{x}_j)\right)\mathbf{e}(\mathbf{x}_i)$, where $\mathbf{x}_i$ denotes different input samples and $\mathbf{e}(\mathbf{x}_i)$ denotes the error for the output layer (i.e., gradient) with input $\mathbf{x}_i$. We show that this gradient can provide a similar descent direction as surrogate gradients under certain conditions, and the proof and discussions can be found in Appendix B. We will also show empirical verifications of our FGT.

**Proposition 1.** *Assume that $\widetilde{\mathbf{J}_{\mathbf{f}}}(\mathbf{x})$ is $L_J$-Liptschitz continuous and $\mathbf{e}(\mathbf{x})$ is $L_e$-Liptschitz continuous, $\mathbf{x}_i$ is uniformly distributed data sample, when $\left\| \mathbb{E}_{\mathbf{x}_i}\widetilde{\mathbf{J}_{\mathbf{f}}}(\mathbf{x}_i)\mathbf{e}(\mathbf{x}_i) \right\| \geq \frac{1}{2}L_J L_e \Delta_{\mathbf{x}}$, where $\Delta_{\mathbf{x}} = \mathbb{E}_{\mathbf{x}_i, \mathbf{x}_j}\|\mathbf{x}_i - \mathbf{x}_j\|^2$, we have $\left\langle \mathbb{E}_{\mathbf{x}_i}\widetilde{\mathbf{J}_{\mathbf{f}}}(\mathbf{x}_i)\mathbf{e}(\mathbf{x}_i), \mathbb{E}_{\mathbf{x}_i}\left(\mathbb{E}_{\mathbf{x}_j}\widetilde{\mathbf{J}_{\mathbf{f}}}(\mathbf{x}_j)\right)\mathbf{e}(\mathbf{x}_i) \right\rangle \geq 0$.*

### 4.3 FORWARD AND ONLINE TRAINING

Building forward gradient training on online training methods, we can obtain forward and online training methods for SNNs. We consider OTTT in this work as introduced in Section 3.2. By replacing the backpropagated instantaneous gradient with forward gradients ($\mathbf{g}_{\mathbf{s}^{l+1}}$ or $\overline{\mathbf{g}_{\mathbf{s}^{l+1}}}$, we use the notation $\mathbf{g}_{\mathbf{s}^{l+1}}^t$ for simplicity and the superscript $t$ means that the gradient corresponds to the forward propagation at the $t$-th time step), the gradients at each time step are calculated by $\nabla_{\mathbf{W}^l} L[t] = (\mathbf{g}_{\mathbf{s}^{l+1}}^t \odot f(\mathbf{u}^{i+1}[t]))\hat{\mathbf{a}}^l[t]^\top$. This also has a similar form as the three-factor Hebbian learning (Frémaux & Gerstner, 2016) with a global modulator directly from the top layer:

$$\nabla_{W_{i,j}} L[t] = \hat{a}_i[t] f(u_j[t]) g_j^t, \tag{8}$$

where $W_{i,j}$ is the weight connecting neuron $i$ and $j$, $\hat{a}_i[t]$ is the tracked presynaptic activity, $f(u_j[t])$ is the surrogate derivative which can represent the change rate of the postsynaptic activity (Xiao et al., 2022), and $g_j^t$ is a global error (gradient) modulator directly from the output layer (Fig. 1(e,f)).

Considering neuromorphic computing, there may be some delay of the error signal (for forward propagation and feedback from the top layer). Xiao et al. (2022) has shown that under the condition of convergent input and a certain surrogate function, the gradient can still be theoretically effective under the delay $\Delta t$, i.e., the update is based on $\hat{a}_i[t+\Delta t] f(u_j[t+\Delta t]) g_j^t$. Our method also shares the conclusion. For simplicity and efficiency of simulation, we do not model the delay in experiments.

### 4.4 DISCUSSION AND IMPLEMENTATION DETAILS

**Signal Propagation of Forward Gradient** The propagation of forward gradients is simultaneous and coupled with the forward propagation, and a natural question is how can the signal be propagated. Actually, Payeur et al. (2021) have shown that signals in neural systems with different frequencies can be simultaneously propagated with different information, and they demonstrate that bursts of spikes may backpropagate errors. So we conjecture that the forward gradient propagation can also be realized by neurons, e.g., with bursts. For example, after a spike in the common forward propagation, neurons may generate bursts with burst inputs that represent forward gradients, and signals are encoded by the rate. To simulate this condition, we consider quantizing signals in Eq. (4), i.e., the output of neurons are $\mathbf{z}^{i+1,l} = Q(\mathbf{W}^i \mathbf{z}^{i,l} \odot f(\mathbf{u}^{i+1}[t]))$ representing the rate of bursts with a few time steps (more details can be found in Appendix D). As will be shown, this works well even under the condition of only one time step, i.e., only one spike for the signal is enough.

Another question is how signals of $\mathbf{z}^{i,l}$ for different layers $l$ are propagated. By default, we sequentially sample a $\mathbf{v}^l$ for the $l$-th layer during the forward propagation and add it to a list for propagation with Eq. (4). To improve the efficiency and biological plausibility, when using momentum feedback connections, we also consider only sampling one layer for forward gradient propagation and feedback connection update in each iteration, so that there will only be one gradient signal in the propagation. Experiments show that it also works well. Additionally, we may also consider pretraining feedback connections with forward gradients unsupervisedly before supervised learning, and update them after a certain interval during training. In experiments, we update feedback connections by forward gradients simultaneously with training. Pseudocodes are in Appendix C.

**Combination with Local Learning** Biological systems can learn from both global and local signals. Our work mainly focuses on global learning with modulation signals directly from the top layer and is compatible with local learning. In experiments, we will show that introducing local learning with local readout can further improve performance. We can add a fully connected readout for each layer to provide local supervision with supervised loss (more details in Appendix D).

## 5 EXPERIMENTS

In this section, we demonstrate the effectiveness of FGT on both fully connected (FC) and convolutional (Conv) neural networks for static and neuromorphic datasets. We conduct experiments on MNIST and N-MNIST with FC networks consisting of two hidden layers with 400 neurons, and experiments on DVS-CIFAR10, DVS-Gesture, CIFAR-10, and CIFAR-100 with 5-layer convolutional networks (128C3-AP2-256C3-AP2-512C3-AP2-512C3-FC). Following previous works (Xiao et al., 2022; Zhang & Li, 2020), we take $T = 6$ time steps for static datasets, $T = 30$ for N-MNIST, $T = 20$ for DVS-Gesture, and $T = 10$ for DVS-CIFAR10. By default, we consider the deterministic neuron model, and we will show results under the stochastic setting later. Please refer to Appendix D for training details.

For our proposed methods, let "FGT" denote the default forward gradient with momentum feedback connections, "FGT (w/o m)" denote vanilla forward gradient, "FGT (w/o m, s8)" denote vanilla forward gradient with 8 samples of $\mathbf{v}$, "FGT (Q20)" / "FGT (Q1)" denote representing signals by bursts with 20 / 1 time steps (Section 4.4), and "FGT (S)" denote only sampling one layer for forward gradient calculation in each iteration (Section 4.4). All results are based on the OTTT method.

Table 1: Accuracy (%) of different FGT methods on MNIST with fully connected networks and DVS-CIFAR10 with convolutional networks.

| Dataset | FGT (w/o m) | FGT (w/o m, s8) | FGT | FGT (Q20) | FGT (Q1) | FGT (S) |
|---|---|---|---|---|---|---|
| MNIST | 96.06±0.12 | 97.62±0.11 | 98.12±0.00 | 98.14±0.00 | 98.22±0.06 | 98.10±0.01 |
| DVS-CIFAR10 | 38.30±0.45 | 49.70±0.98 | 73.30±0.08 | 73.40±0.16 | 73.30±0.24 | 72.90± 0.36 |

Table 2: Accuracy (%) of different spatial credit assignment methods on MNIST with fully connected networks and DVS-CIFAR10 with convolutional networks. LL denotes local learning.

| Dataset | BP | DFA | LL | DFA + LL | FGT | FGT + LL |
|---|---|---|---|---|---|---|
| MNIST | 98.19±0.05 | 97.83±0.03 | / | / | 98.12±0.00 | / |
| DVS-CIFAR10 | 75.57±0.57 | 60.20±0.42 | 48.53±0.75 | 61.70±0.36 | 73.30±0.08 | 75.13±0.46 |

**Results of different forward propagation training methods** As shown in Table 1, vanilla forward gradient training cannot work well, especially for convolutional networks with a larger number of neurons. Simply sampling a few more directions for variance reduction can slightly improve the performance with larger training costs, but still cannot reach the competitive level. Also, note that we train models by the Adam optimizer, so momentum techniques in the optimizer alone cannot solve the problem. On the other hand, forward gradients with momentum feedback connections can significantly improve the results, achieving similar results as BP (Table 2). Meanwhile, representing signals by the rate of bursts with a few time steps can achieve similar performance, even with only one time step (i.e., one spike). FGT with the signal of only one sampled layer per iteration also works well. These results demonstrate the effectiveness and robustness of the proposed FGT.

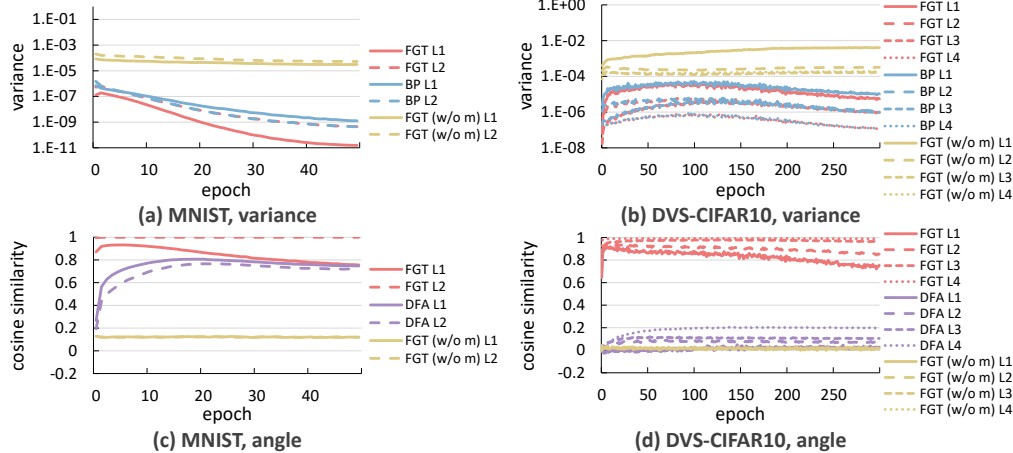

Figure 2: Results of gradient variances of different methods and gradient angles (cosine similarity) between different methods and backpropagated gradients. "Li" denotes the $i$-th layer.

**Comparison with other spatial credit assignment methods** Table 2 summarizes the results of different spatial credit assignment methods including BP, DFA, FGT, and combination with local learning. For FC networks, since there are only two hidden layers, we do not consider local learning settings. Results show that there is a large gap between DFA or local learning and BP, especially for convolutional networks, while FGT can achieve more competitive performance. When combined with local learning, FGT can achieve similar results as BP. Table 3 presents more results on different datasets. The results show the effectiveness of FGT in achieving similar performance as BP under various settings while avoiding the drawbacks of BP including layer-by-layer forward-backward symmetric weights and separate phases, and FGT (Q1, S) provides a more efficient and neuromorphic-friendly alternative. This can pave paths for on-chip training of SNNs.

Table 3: More accuracy (%) results on N-MNIST with fully connected networks as well as DVS-Gesture, CIFAR-10, and CIFAR-100 with convolutional networks. LL denotes local learning.

| Dataset | BP | DFA | FGT (Q1, S) | FGT | FGT + LL |
|---|---|---|---|---|---|
| N-MNIST | 97.77±0.03 | 97.68±0.02 | 97.93±0.04 | 97.98±0.00 | / |
| DVS-Gesture | 97.22±0.28 | 82.87±0.43 | 94.44±0.57 | 95.60±0.43 | 96.30±0.16 |
| CIFAR-10 | 89.29±0.24 | 79.92±0.13 | 85.69±0.33 | 85.98±0.15 | 86.98±0.23 |
| CIFAR-100 | 63.51±0.11 | 50.09±0.56 | 60.81±0.21 | 61.19±0.26 | 62.16±0.14 |

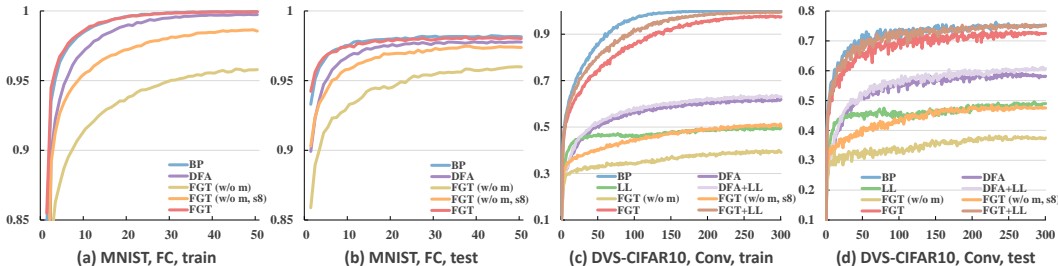

Figure 3: Training dynamics (accuracy w.r.t epoch) of different spatial credit assignment methods.

Table 4: Accuracy (%) of different methods with the stochastic neuron model on various datasets.

| Method | MNIST | N-MNIST | DVS-Gesture | DVS-CIFAR10 | CIFAR-10 | CIFAR-100 |
|---|---|---|---|---|---|---|
| BP | 98.23±0.04 | 97.91±0.02 | 97.57±0.00 | 75.17±0.56 | 89.43±0.07 | 63.36±0.25 |
| FGT | 98.10±0.09 | 97.88±0.02 | 95.02±0.33 | 73.20±0.14 | 86.17±0.27 | 63.99±0.24 |
| FGT + LL | / | / | 96.99±0.16 | 75.17±0.12 | 87.17±0.16 | 64.60±0.24 |

**Gradient variance and angles** We conduct experiments of gradient variance and angles to verify that our momentum feedback connections can effectively reduce variance and maintain valid descent directions. As shown in Fig. 2, FGT can effectively reduce the variance of the vanilla forward gradient by several orders, leading to a similar variance as BP, and the average cosine similarity between FGT gradient and BP gradient is always above 0.75 throughout training, while DFA has much worse results for convolutional networks (always below 0.2). The vanilla forward gradient is close to orthogonal due to the projection to a random direction with high dimensions.

**Training dynamics** Fig. 3 illustrates the training dynamics of different methods. It shows that the convergence speed of FGT for fully connected networks is faster than DFA and comparable to BP. For convolutional networks, FGT is slower than BP but performs much better than DFA and local learning. Meanwhile, local learning can improve FGT.

**Effectiveness for stochastic neuron models** As described in Section 3.1, stochastic neuron models have clearer grounding for surrogate gradients and may better correspond to hardware noise. Our method is also applicable to the stochastic setting. As shown in Table 4, FGT generally achieves similar or better results under the stochastic setting compared with the deterministic condition, especially on CIFAR-100 with $\geq 2.44\%$ accuracy improvement, while BP has similar performance.

More results and discussions of training costs, firing rate statistics, deeper networks, and finetuning ResNet-34 on ImageNet under noise are in Appendix E.

## 6 CONCLUSION

In this work, we propose a new training method, forward gradient training (FGT), for spiking neural networks. FGT enables spatial credit assignment with only unidirectional forward propagation across layers and direct feedback from the top layer, avoiding drawbacks of BP across layers while achieving similar performance. We show that FGT can be combined with online training methods for forward and online training of SNNs. This takes a step forward towards on-chip SNN training with biologically more plausible methods. FGT also provides the possibility for feedback signals directly from the top layer with clearer meaning and more theoretical guarantees instead of random feedback. Extensive experiments demonstrate the effectiveness and robustness of the proposed method for both fully connected and convolutional networks under static and neuromorphic datasets.

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

# A    MORE INTRODUCTION TO STOCHASTIC SPIKING NEURON MODELS AND SURROGATE GRADIENTS

As introduced in Section 3.1, the stochastic spiking neuron model generates spikes with a probability distribution. The deterministic model generates spikes once the membrane potential exceeds the threshold:

$$s_i[t] = H(u_i[t] - V_{th}), \tag{9}$$

while for the stochastic model, $s_i[t]$ follows $\{0, 1\}$ valued Bernoulli distribution with probability $p(s_i[t] = 1) = F(u_i[t] - V_{th})$, where $F$ is the cdf of the distribution. This can be realized by reparameterization, i.e., the spike generation is:

$$s_i[t] = H(u_i[t] - V_{th} - z_i), \tag{10}$$

where $z_i$ is a random variable following the distribution of $F$. Such formulation can be viewed as a noise injection for the membrane potential, which may correspond to some hardware noise, e.g., thermal noise (Büchel et al., 2021), for neuromorphic computing. Different $F$ corresponds to different noises. For example, for the sigmoid function, the noise corresponds to the logistic noise, while for the erf function, the noise corresponds to the Gaussian noise. The commonly used surrogate functions all have corresponding noises (Shekhovtsov & Yanush, 2021).

For the stochastic model, the objective function should be changed to the expectation over the random variables. Specifically, we first consider a single hidden layer network with a single time step for simplicity. The input $\mathbf{x}$ are connected to $n$ hidden spiking neurons by the weight $\mathbf{W}$, which are then connected to an output readout layer by the weight $\mathbf{O}$. For deterministic models, the objective function is to minimize $\mathbb{E}_{\mathbf{x}}[\mathcal{L}(\mathbf{s})]$, with $\mathbf{s} = H(\mathbf{u} - V_{th}), \mathbf{u} = \mathbf{Wx}$. For stochastic models, the objective is to minimize

$$\mathbb{E}_{\mathbf{x}}[\mathbb{E}_{\mathbf{s} \sim p(\mathbf{s}|\mathbf{x}, \mathbf{W})}[\mathcal{L}(\mathbf{s})]] \tag{11}$$

considering the expectation of $\mathbf{s}$ with the probability distribution whose cdf is $F(\mathbf{u} - V_{th})$ (variables $s_i$ are independent of each other). In this setting, the gradients are actually well-defined, and Shekhovtsov & Yanush (2021); Shekhovtsov et al. (2020) show that a similar form of surrogate gradients (also known as straight-through-estimator) can be derived through derandomization and linear approximation. Specifically, we focus on how gradients of $\mathbf{u}$ can be derived. Given inputs, the gradients are expressed as:

$$\begin{aligned}
\frac{\partial}{\partial \mathbf{u}} \mathbb{E}_{\mathbf{s} \sim p(\mathbf{s}|\mathbf{W})}[\mathcal{L}(\mathbf{s})] &= \frac{\partial}{\partial \mathbf{u}} \sum_{\mathbf{s}} \left( \prod_i p(\mathbf{s}_i|\mathbf{W}) \right) \mathcal{L}(\mathbf{s}) \\
&= \sum_{\mathbf{s}} \sum_i \left( \prod_{i' \neq i} p(\mathbf{s}_{i'}|\mathbf{W}) \right) \left( \frac{\partial}{\partial \mathbf{u}} p(\mathbf{s}_i|\mathbf{W}) \right) \mathcal{L}(\mathbf{s}).
\end{aligned} \tag{12}$$

Then derandomization can be applied, i.e., performing summation over $s_i$ with other random variables fixed (Shekhovtsov & Yanush, 2021). Let $\mathbf{s}_{\neg i}$ denote all variables excluding $s_i$. Since $s_i$ is binary valued, given $\mathbf{s}_{\neg i}$, we have

$$\begin{aligned}
\sum_{s_i \in \{0,1\}} \frac{\partial p(s_i|\mathbf{W})}{\partial \mathbf{u}} \mathcal{L}([\mathbf{s}_{\neg i}, s_i]) &= \frac{\partial p(\mathbf{s}_i|\mathbf{W})}{\partial \mathbf{u}} \mathcal{L}(\mathbf{s}) + \frac{\partial(1 - p(\mathbf{s}_i|\mathbf{W}))}{\partial \mathbf{u}} \mathcal{L}(\mathbf{s}_{\downarrow i}) \\
&= \frac{\partial p(\mathbf{s}_i|\mathbf{W})}{\partial \mathbf{u}} \left( \mathcal{L}(\mathbf{s}) - \mathcal{L}(\mathbf{s}_{\downarrow i}) \right),
\end{aligned} \tag{13}$$

where $[\mathbf{s}_{\neg i}, s_i]$ denotes taking the values of $\mathbf{s}_{\neg i}$ and $s_i$ for random variables, $\mathbf{s}$ is a sample ($s_i$ can be either 0 or 1, i.e., the RHS is invariant of $s_i$), and $\mathbf{s}_{\downarrow i}$ denotes that $\mathbf{s}_i$ is changed to the other state

while other variables are fixed. Since $1 = \sum_{s_i} p(s_i|\mathbf{W})$, Eq. (12) can be written as

$$
\begin{aligned}
\frac{\partial}{\partial \mathbf{u}} \mathbb{E}_{\mathbf{s} \sim p(\mathbf{s}|\mathbf{W})}[\mathcal{L}(\mathbf{s})] &= \sum_i \sum_{\mathbf{s}_{\neg i}} \left( \prod_{i' \neq i} p(\mathbf{s}_{i'}|\mathbf{W}) \right) \sum_{s_i} \left( \frac{\partial}{\partial \mathbf{u}} p(s_i|\mathbf{W}) \right) \mathcal{L}([\mathbf{s}_{\neg i}, s_i]) \\
&= \sum_i \sum_{\mathbf{s}_{\neg i}} \left( \prod_{i' \neq i} p(\mathbf{s}_{i'}|\mathbf{W}) \right) \sum_{s_i} p(s_i|\mathbf{W}) \frac{\partial p(\mathbf{s}_i|\mathbf{W})}{\partial \mathbf{u}} (\mathcal{L}(\mathbf{s}) - \mathcal{L}(\mathbf{s}_{\downarrow i})) \\
&= \sum_{\mathbf{s}} \left( \prod_i p(\mathbf{s}_i|\mathbf{W}) \right) \sum_i \frac{\partial p(\mathbf{s}_i|\mathbf{W})}{\partial \mathbf{u}} (\mathcal{L}(\mathbf{s}) - \mathcal{L}(\mathbf{s}_{\downarrow i})) \\
&= \mathbb{E}_{\mathbf{s} \sim p(\mathbf{s}|\mathbf{W})} \sum_i \frac{\partial p(\mathbf{s}_i|\mathbf{W})}{\partial \mathbf{u}} (\mathcal{L}(\mathbf{s}) - \mathcal{L}(\mathbf{s}_{\downarrow i})).
\end{aligned}
\tag{14}
$$

We can take one sample in each forward procedure for the unbiased gradient estimation as the Monte Carlo method. Considering the probability distribution, the term $\frac{\partial p(\mathbf{s}_i|\mathbf{W})}{\partial \mathbf{u}}$ just correspond to a surrogate function $f(\mathbf{u}, V_{th})$, which is the derivative of the cdf function $F$, e.g., the derivative of the sigmoid function, triangular function, etc. The term $\mathcal{L}(\mathbf{s}) - \mathcal{L}(\mathbf{s}_{\downarrow i})$ can be tackled by linear approximation considering that $\mathbf{s}_i$ is $\{0, 1\}$ valued: $\mathcal{L}(\mathbf{s}) - \mathcal{L}(\mathbf{s}_{\downarrow i}) \approx \frac{\partial \mathcal{L}(\mathbf{s})}{\partial \mathbf{s}_i}$, given the gradient $\frac{\partial \mathcal{L}(\mathbf{s})}{\partial \mathbf{s}}$. Then the surrogate gradient is systematically derived:

$$
\frac{\widetilde{\partial \mathcal{L}(\mathbf{s})}}{\partial \mathbf{u}} = \frac{\partial \mathcal{L}(\mathbf{s})}{\partial \mathbf{s}} \frac{\widetilde{\partial \mathbf{s}}}{\partial \mathbf{u}}, \quad \frac{\widetilde{\partial \mathbf{s}}}{\partial \mathbf{u}} = \frac{\partial p(\mathbf{s}_i|\mathbf{W})}{\partial \mathbf{u}} = f(\mathbf{u}, V_{th}).
\tag{15}
$$

The linear approximation can introduce bias, while it can be reduced considering over-parameterized neural networks whose weights are at the scale of $\frac{1}{\sqrt{m}}$, where $m$ is the neuron number — for each element of the readout $\mathbf{o} = \mathbf{Os}$, changing the state of one $\mathbf{s}_i$ only has $O(\frac{1}{\sqrt{m}})$ influence. Shekhovtsov & Yanush (2021) shows that the gradient provides a valid ascent direction under a certain condition considering the gradient scale and Lipschitz constant of the gradient. At the same time, the deterministic model can be viewed as a special case, e.g., the samples of noise are always taken as zero, and Shekhovtsov & Yanush (2021) also shows that the deterministic gradient gives a similar ascent direction under certain conditions considering the gradient scale, the Lipchitz constant of the gradient, and the probability for stochastic neurons to generate the same spikes. For more details, please refer to Shekhovtsov & Yanush (2021).

For multi-layer networks, the results can be similarly derived by iteratively performing the above analysis (Shekhovtsov & Yanush, 2021), and it can be the same for gradients at each time step. In practice, surrogate gradients also achieve successful results. Therefore, we consider that surrogate gradients can be reliable targets, and our forward gradients are to estimate them.

## B  DETAILED PROOFS

### B.1  PROOF OF PROPERTY 1

*Proof.* For the forward gradient $\mathbf{g}_{\mathbf{s}^{l+1}} = \left\langle \mathbf{e}^N, \mathbf{z}^{N,l+1} \right\rangle \mathbf{v}^{l+1}$, according to the definition of the surrogate gradient and $\mathbf{z}^{N,l+1}$, we have

$$
\left\langle \mathbf{e}^N, \mathbf{z}^{N,l+1} \right\rangle = \left\langle \widetilde{\nabla_{\mathbf{s}^{l+1}} \mathcal{L}}[t], \mathbf{v}^{l+1} \right\rangle = \sum_j \left( \widetilde{\nabla_{\mathbf{s}^{l+1}} \mathcal{L}}[t] \right)_j \mathbf{v}_j^{l+1}.
\tag{16}
$$

Then for each element of $(\mathbf{g}_{\mathbf{s}^{l+1}})_i$, we have

$$
(\mathbf{g}_{\mathbf{s}^{l+1}})_i = \left( \widetilde{\nabla_{\mathbf{s}^{l+1}} \mathcal{L}}[t] \right)_i \mathbf{v}_i^{l+1^2} + \sum_{j \neq i} \left( \widetilde{\nabla_{\mathbf{s}^{l+1}} \mathcal{L}}[t] \right)_j \mathbf{v}_i^{l+1} \mathbf{v}_j^{l+1}.
\tag{17}
$$

Since $\mathbf{v}^{l+1}$ has i.i.d. components with zero mean and unit variance, we have $\mathbb{E}\left[ \mathbf{v}_i^{l+1^2} \right] = 1$ and $\mathbb{E}\left[ \mathbf{v}_i^{l+1} \mathbf{v}_j^{l+1} \right] = 0$. Then the expectation of each element is

$$
\mathbb{E}\left[ (\mathbf{g}_{\mathbf{s}^{l+1}})_i \right] = \left( \widetilde{\nabla_{\mathbf{s}^{l+1}} \mathcal{L}}[t] \right)_i \mathbb{E}\left[ \mathbf{v}_i^{l+1^2} \right] + \sum_{j \neq i} \left( \widetilde{\nabla_{\mathbf{s}^{l+1}} \mathcal{L}}[t] \right)_j \mathbb{E}\left[ \mathbf{v}_i^{l+1} \mathbf{v}_j^{l+1} \right] = \left( \widetilde{\nabla_{\mathbf{s}^{l+1}} \mathcal{L}}[t] \right)_i.
\tag{18}
$$

Therefore, $\mathbb{E}[\mathbf{g}_{\mathbf{s}^{l+1}}] = \widetilde{\nabla_{\mathbf{s}^{l+1}}\mathcal{L}}[t]$. $\qquad\qquad\qquad\qquad\qquad\qquad\qquad\qquad\qquad\qquad\square$

## B.2 PROOF OF PROPOSITION 1

*Proof.* According to the Lipschitz condition, it holds that $\left\|\widetilde{\mathbf{J}_{\mathbf{f}}}(\mathbf{x}_i) - \widetilde{\mathbf{J}_{\mathbf{f}}}(\mathbf{x}_j)\right\| \leq L_J \|\mathbf{x}_i - \mathbf{x}_j\|$, $\|\mathbf{e}(\mathbf{x}_i) - \mathbf{e}(\mathbf{x}_j)\| \leq L_e \|\mathbf{x}_i - \mathbf{x}_j\|$. Then based on the equation that $\frac{1}{2n^2}\sum_{i,j}(a_i - a_j)(b_i - b_j) = \frac{1}{n}\sum_i a_i b_i - \frac{1}{n^2}\sum_{i,j} a_i b_j$, we have

$$
\begin{aligned}
&\left\|\mathbb{E}_{\mathbf{x}_i}\widetilde{\mathbf{J}_{\mathbf{f}}}(\mathbf{x}_i)\mathbf{e}(\mathbf{x}_i) - \mathbb{E}_{\mathbf{x}_i}\left(\mathbb{E}_{\mathbf{x}_j}\widetilde{\mathbf{J}_{\mathbf{f}}}(\mathbf{x}_j)\right)\mathbf{e}(\mathbf{x}_i)\right\| \\
=\ & \left\|\frac{1}{n}\sum_{\mathbf{x}_i}\widetilde{\mathbf{J}_{\mathbf{f}}}(\mathbf{x}_i)\mathbf{e}(\mathbf{x}_i) - \left(\frac{1}{n}\sum_{\mathbf{x}_i}\widetilde{\mathbf{J}_{\mathbf{f}}}(\mathbf{x}_i)\right)\left(\frac{1}{n}\sum_{\mathbf{x}_i}\mathbf{e}(\mathbf{x}_i)\right)\right\| \\
=\ & \left\|\frac{1}{2n^2}\sum_{\mathbf{x}_i,\mathbf{x}_j}\left(\widetilde{\mathbf{J}_{\mathbf{f}}}(\mathbf{x}_i) - \widetilde{\mathbf{J}_{\mathbf{f}}}(\mathbf{x}_j)\right)(\mathbf{e}(\mathbf{x}_i) - \mathbf{e}(\mathbf{x}_j))\right\| \\
\leq\ & \frac{1}{2n^2}\sum_{\mathbf{x}_i,\mathbf{x}_j}\left\|\left(\widetilde{\mathbf{J}_{\mathbf{f}}}(\mathbf{x}_i) - \widetilde{\mathbf{J}_{\mathbf{f}}}(\mathbf{x}_j)\right)\right\|\|(\mathbf{e}(\mathbf{x}_i) - \mathbf{e}(\mathbf{x}_j))\| \\
\leq\ & \frac{1}{2n^2}\sum_{\mathbf{x}_i,\mathbf{x}_j}L_J L_e\|\mathbf{x}_i - \mathbf{x}_j\|^2 = \frac{1}{2}L_J L_e \Delta_{\mathbf{x}} \\
\leq\ & \left\|\mathbb{E}_{\mathbf{x}_i}\widetilde{\mathbf{J}_{\mathbf{f}}}(\mathbf{x}_i)\mathbf{e}(\mathbf{x}_i)\right\|.
\end{aligned}
\tag{19}
$$

Therefore,

$$
\begin{aligned}
&\left\langle\mathbb{E}_{\mathbf{x}_i}\widetilde{\mathbf{J}_{\mathbf{f}}}(\mathbf{x}_i)\mathbf{e}(\mathbf{x}_i), \mathbb{E}_{\mathbf{x}_i}\left(\mathbb{E}_{\mathbf{x}_j}\widetilde{\mathbf{J}_{\mathbf{f}}}(\mathbf{x}_j)\right)\mathbf{e}(\mathbf{x}_i)\right\rangle \\
=\ & \left\|\mathbb{E}_{\mathbf{x}_i}\widetilde{\mathbf{J}_{\mathbf{f}}}(\mathbf{x}_i)\mathbf{e}(\mathbf{x}_i)\right\|^2 - \left\langle\mathbb{E}_{\mathbf{x}_i}\widetilde{\mathbf{J}_{\mathbf{f}}}(\mathbf{x}_i)\mathbf{e}(\mathbf{x}_i), \mathbb{E}_{\mathbf{x}_i}\widetilde{\mathbf{J}_{\mathbf{f}}}(\mathbf{x}_i)\mathbf{e}(\mathbf{x}_i) - \mathbb{E}_{\mathbf{x}_i}\left(\mathbb{E}_{\mathbf{x}_j}\widetilde{\mathbf{J}_{\mathbf{f}}}(\mathbf{x}_j)\right)\mathbf{e}(\mathbf{x}_i)\right\rangle \\
\geq\ & \left\|\mathbb{E}_{\mathbf{x}_i}\widetilde{\mathbf{J}_{\mathbf{f}}}(\mathbf{x}_i)\mathbf{e}(\mathbf{x}_i)\right\|^2 - \left\|\mathbb{E}_{\mathbf{x}_i}\widetilde{\mathbf{J}_{\mathbf{f}}}(\mathbf{x}_i)\mathbf{e}(\mathbf{x}_i)\right\|\left\|\mathbb{E}_{\mathbf{x}_i}\widetilde{\mathbf{J}_{\mathbf{f}}}(\mathbf{x}_i)\mathbf{e}(\mathbf{x}_i) - \mathbb{E}_{\mathbf{x}_i}\left(\mathbb{E}_{\mathbf{x}_j}\widetilde{\mathbf{J}_{\mathbf{f}}}(\mathbf{x}_j)\right)\mathbf{e}(\mathbf{x}_i)\right\| \\
\geq\ & 0.
\end{aligned}
\tag{20}
$$

$\square$

**Remark 1.** *The proposition makes assumptions on the Liptschitz condition of the surrogate Jacobian and the gradient of the output layer w.r.t. inputs. This can be reasonable under the stochastic neuron setting where surrogate gradients are well-defined and objective functions are the expectation over random variables. $L_J$ will depend on the non-linearity of networks, for example, $L_J = 0$ for linear networks. This will influence the condition of effective descent direction considering the gradient scale as in the proposition. Actually, these assumptions are not necessary premises, and FGT with momentum feedback connections is a practical approximation to the gradient considered in the proposition. We have empirically shown that FGT provides effective descent directions. This proposition mainly provides insights into the condition with the expectation of (surrogate) Jacobian that FGT approximates.*

## C PSEUDOCODE OF THE FGT ALGORITHM

We present the pseudocode of one iteration of FGT training with momentum feedback connections in Algorithm 1 to better illustrate our training method. It is based on the OTTT method (Xiao et al., 2022), and we present $\text{OTTT}_A$ with the accumulation of several time steps for simplicity, while it is also possible for the online $\text{OTTT}_O$.

---

**Algorithm 1** One iteration of FGT training for a feedforward network.

---

**Input:** Network parameters $\{\mathbf{W}^l\}$; Feedback connections $\{\mathbf{M}^l\}$; Input data $x$; Label $y$; Time steps $T$; Other hyperparameters;
**Output:** Trained network parameters $\{\mathbf{W}^l\}$, updated $\{\mathbf{M}^l\}$.

---

1: **for** $t = 1, 2, \cdots, T$ **do**
2:     z_list = [].    // The list of forward gradients for different layers, we use z to distinguish them from sampled $\mathbf{v}^l$ at each layer
3:     **for** $l = 1, 2, \cdots, N$ **do**    // **Forward propagation**
4:         Update membrane potentials $\mathbf{u}^l[t]$ and generate spikes $\mathbf{s}^l[t]$ at layer $l$;
5:         Update the tracked presynaptic activities $\hat{\mathbf{a}}^l[t] = \lambda\hat{\mathbf{a}}^l[t-1] + \hat{\mathbf{s}}^l[t]$ at layer $l$;
6:         Forward propagate elements in z_list as Eq. (4);
7:         Sample $\mathbf{v}^l$, z_list.append($\mathbf{v}^l$).    // For FGT (S), only one random layer will sample $\mathbf{v}^l$ for forward propagation
8:     Get local gradient $\mathbf{e}^N$ at the output layer.
9:     **for** $l = 1, 2, \cdots, N-1$ **do**    // **Feedback from the top layer, parallelizable**
10:         Update $\mathbf{M}^l$ as Eq. (6) based on $\mathbf{v}^l$ and elements in z_list at the output layer;    // For FGT (S), only one random layer will update
11:         Propagate errors from the top layer to obtain feedback errors $\mathbf{g}_{\mathbf{s}^l}[t] = \mathbf{M}^l\mathbf{e}^N$;
12:         Calculate the instantaneous gradient $\nabla_{\mathbf{W}^l}L[t]$ based on $\mathbf{g}_{\mathbf{s}^{l+1}}[t]$ and $\hat{\mathbf{a}}^l[t]$;
13:         Accumulate gradients $\nabla_{\mathbf{W}^l}L = \nabla_{\mathbf{W}^l}L + \nabla_{\mathbf{W}^l}L[t]$.
14: Update parameters $\{\mathbf{W}^l\}$ with accumulated $\{\nabla_{\mathbf{W}^l}L\}$ based on the gradient-based optimizer.

---

# D   IMPLEMENTATION DETAILS

## D.1   ENCODING SIGNALS BY THE RATE OF BURSTS

As introduced in Section 4.4, we will consider representing signals of forward gradients by the rate of bursts. We consider $T_B$ discrete time steps for the bursts and take the spiking rate as the signal. Specifically, given an input signal $x$, a spike train with $T_B$ discrete time steps is outputted which can also be viewed as the output of the non-leaky integrate and fire neuron with the constant input $x$, and the rate of the spike train is regarded as the signal. Then given a scale $s$ for each spike, this can correspond to a quantization of the input signal $x$ as $Q(x) = s \times \frac{\left[\frac{\text{clamp}(x, -s, s)}{s} \times T_B\right]}{T_B}$. We use this formulation in experiments and for $T_B = 20$, we take $s = 10$, while for $T_B = 1$, we take $s = 1$. Note that the discretization step size for the bursts can have a different scale than the common forward propagation, i.e., they are high-frequency signals and multiple discrete steps of bursts can correspond to one step of the common forward signals. Also note that the signals may require negative spikes to propagate negative inputs, i.e., neurons will generate a negative spike when the membrane potential is lower than a negative threshold. This is inevitable for the propagation of gradient signals which can be either positive or negative, and is shared by methods to propagate gradient signals across layers. One solution is to enable negative spiking. Another solution for common positively spiking neuron models can be leveraging a neuron couple with recurrent connections between each other to simulate ternary spikes, as illustrated in Xiao et al. (2023) (more details please refer to Xiao et al. (2023)).

## D.2   COMBINATION WITH LOCAL LEARNING

In our experiments of local learning, we simply consider fully connected readout layers for each layer to provide supervision. Specifically, for the output $\mathbf{s}^l$ of each layer, we obtain the readout $\mathbf{r}^l = \mathbf{R}^l\mathbf{s}^l$ to calculate loss based on it: $\mathcal{L}(\mathbf{r}^l, \mathbf{y})$. Then the gradient for $\mathbf{s}^l$ is calculated by the local loss, and it is only leveraged to update synaptic weights directly connected to the neurons. Note that for simplicity, we assume the weight symmetry for propagating errors through $\mathbf{R}^l$ in the experiments, and the readout weight is learned with local errors, while Kaiser et al. (2020) has shown that fixed random matrix and sign symmetry can be effective for such local learning without weight symmetry. It can be extended to these settings, as well as other local learning methods.

### D.3 TRAINING SETTINGS

#### D.3.1 DATASETS

We conduct experiments on MNIST (LeCun et al., 1998), N-MNIST (Orchard et al., 2015), DVS-CIFAR10 (Li et al., 2017), DVS-Gesture (Amir et al., 2017), CIFAR-10 (Krizhevsky & Hinton, 2009), and CIFAR-100 (Krizhevsky & Hinton, 2009).

**MNIST** The MNIST dataset consists of 10-class handwritten digits with 60,000 training samples and 10,000 testing samples. Each sample is a $28 \times 28$ grayscale image. We normalize the inputs based on the global mean and standard deviation, and convert the pixel value into a real-valued input current at every time step. No data augmentation is applied. The license of MNIST is the MIT License.

**N-MNIST** The N-MNIST dataset is a neuromorphic dataset converted from MNIST by a Dynamic Version Sensor (DVS). It consists of spike trains triggered by the intensity change of pixels when DVS scans the static MNIST images along given directions. Since the intensity can either increase or decrease, there are two channels corresponding to ON- and OFF-event spikes. And the pixel dimension is expanded to $34 \times 34$ due to the relative shift of images. Therefore, each sample is a spike train pattern with the size of $34 \times 34 \times 2 \times T$, where $T$ is the temporal length. The original data record $300ms$ with the resolution of $1\mu s$. We follow the prepossessing of Zhang & Li (2020) to reduce the time resolution by accumulating the spike train within every $3ms$, and we will use the first 30 time steps. The license of N-MNIST is the Creative Commons Attribution-ShareAlike 4.0 license.

**DVS-CIFAR10** The DVS-CIFAR10 dataset is the neuromorphic dataset converted from the CIFAR-10 dataset by DVS, which is composed of 10,000 samples, one-sixth of the original CIFAR-10. It consists of spike trains with two channels corresponding to ON- and OFF-event spikes. The pixel dimension is expanded to $128 \times 128$. Following the common practice, we split the dataset into 9000 training samples and 1000 testing samples. As for the data pre-processing, we reduce the time resolution by accumulating the spike events (Fang et al., 2021) into 10 time steps, and we reduce the spatial resolution into $48 \times 48$ by interpolation. We apply the random cropping augmentation as CIFAR-10 to the input data and normalize the inputs based on the global mean and standard deviation of all time steps (which can be integrated into the connection weights of the first layer). The license of DVS-CIFAR10 is CC BY 4.0.

**DVS-Gesture** The DVS-Gesture dataset is a neuromorphic dataset with 11 kinds of hand gestures from 29 subjects under 3 kinds of illumination conditions recorded by a DVS camera. It is composed of 1176 training samples and 288 testing samples. Following Fang et al. (2021), we pre-possess the data to integrate event data into 20 frames. The license of DVS-Gesture is the Creative Commons Attribution 4.0 license.

**CIFAR-10** The CIFAR-10 dataset consists of 10-class color images of objects, which contains 50,000 training samples and 10,000 testing samples. Each sample is a $32 \times 32 \times 3$ color image. We normalize the inputs based on the global mean and standard deviation, and apply random cropping, horizontal flipping, and cutout (DeVries & Taylor, 2017) for data augmentation. The inputs to the first layer of SNNs at each time step are directly the pixel values, which can be viewed as a real-valued input current.

**CIFAR-100** CIFAR-100 is a dataset similar to CIFAR-10 except that there are 100 classes of objects. It also consists of 50,000 training samples and 10,000 testing samples. We use the same pre-processing as CIFAR-10.

The license of CIFAR-10 and CIFAR-100 is the MIT License.

#### D.3.2 TRAINING DETAILS AND HYPERPARAMETERS

Our training settings mainly follow Xiao et al. (2022). For SNN models, following the common practice, we assume that the neurons of the last classification layer will not spike or reset, and do

classification based on the accumulated membrane potential. That is, the final output is $\mathbf{u}^N[t] = \mathbf{W}^{N-1}\mathbf{s}^{N-1}[t] + \mathbf{b}^N$ at each time step. The classification is based on the accumulated $\mathbf{u}^N = \sum_{t=1}^T \mathbf{u}^N[t]$, and the loss during training is also calculated based on $\mathbf{u}^N[t]$, i.e., $\mathcal{L}(\mathbf{u}^N[t], \mathbf{y})$. And our loss function takes the same form as Xiao et al. (2022), i.e., it is based on the combination of cross-entropy (CE) loss and mean-square-error (MSE) loss. The hyperparameters for spiking neurons are set as $V_{th} = 1$ and $\lambda = 0.5$. We take the sigmoid-like surrogate function with the hyperparameter as $a_1 = 0.25$, and for the stochastic neuron model, the noise corresponds to the logistic noise of this sigmoid function. For convolutional networks, we apply the scaled weight standardization (Brock et al., 2021a;b) as in Xiao et al. (2022).

We train all our models by the AdamW optimizer (Loshchilov & Hutter, 2019) with learning 2e-4 and weight decay 2e-4. The batch size is set as 128 for most datasets and 16 for DVS-Gesture following Xiao et al. (2022), and the learning rate is cosine annealing to 0. For MNIST and N-MNIST, we train models by 50 epochs and we apply dropout with the rate 0.2 (except for N-MNIST with the stochastic model). For DVS-CIFAR10, DVS-Gesture, CIFAR-10, and CIFAR-100, we train models by 300 epochs. For DVS-CIFAR10, we apply dropout with the rate 0.1. We set the momentum coefficient for momentum feedback connections as $\lambda = 0.999$, and for the combination with local learning, the local loss is scaled by 0.01.

The code implementation is based on the PyTorch framework, and experiments are carried out on one NVIDIA GeForce RTX 3090 GPU. All experiments are based on 3 runs of experiments and are carried out under the same random seeds 2022, 0, and 1.

As for gradient variance and angle experiments, the gradient angles are represented by cosine similarity (average per epoch) with the backpropagated gradient and we study FGT, DFA, and FGT (w/o m) (i.e., the vanilla forward gradient), and the gradient variances are calculated by the batch gradients in one epoch, i.e., $var = \frac{\sum \|\mathbf{g}_i - \overline{\mathbf{g}}\|^2}{n}$, where $\mathbf{g}_i$ is the batch gradient, $\overline{\mathbf{g}}$ is the average of batch gradients, and $n$ is the number of batches multiplied by the number of elements in the gradient vector. The variance consists of sample variance as well as additional variance, e.g., introduced by the vanilla forward gradient. We study the variance of FGT, BP, and FGT (w/o m).

# E   ADDITIONAL RESULTS

## E.1   TRAINING COSTS

Table 5: Estimation of training costs on potential neuromorphic hardware. For illustration, we consider neural networks with $N$ hidden layer with $n$ neurons for each layer and $m$ neurons for the output layer, where $m \ll n$. The costs mainly focus on additional synaptic costs besides the normal forward procedure.

| Method | Memory | Operations |
|---|---|---|
| BP (if possible) | $O\left((N-1)n^2 + mn\right)$ | $O\left((N-1)n^2 + mn\right)$ |
| DFA | $O\left(Nmn\right)$ | $O\left(Nmn\right)$ |
| FGT (S) | $O\left(Nmn\right)$ | $O\left(\frac{1}{2}(N-1)n^2 + mn + Nmn\right)$ |

Here we provide a detailed discussion of computational overhead on both common hardware and potential neuromorphic hardware. The theoretical estimation of training costs on neuromorphic hardware is presented in Table 5, while the implemented training costs on GPU are shown in Table 6. Note that neuromorphic hardware is expected to have different architectures than GPU, so the costs can be different. And since we do not fully optimize the codes for low-level optimization of our method as BP in PyTorch, the results on GPU are brief comparisons. It can also be further improved, e.g., with parallelization of computation for forward gradients.

First, for memory cost, let's first consider the cost on CPU/GPU. The symmetric weight of BP does not require additional storage on GPU. For FGT, FGT (S), and DFA, they maintain feedback connections directly from the last layer to middle ones, which can be additional to BP that uses symmetric weights to backpropagate errors. And FGT additionally stores random vectors for $N$

Table 6: Brief comparison of training costs on GPU for CIFAR-100 with convolutional networks. LL denotes local learning.

| Method | Memory | Time per epoch |
|---|---|---|
| BP | 3.2G | 52s |
| DFA | 3.3G | 48s |
| FGT (w/o m) | 3.4G | 67s |
| FGT | 3.4G | 70s |
| FGT (S) | 3.4G | 53s |
| FGT + LL | 3.8G | 76s |

layers while FGT (S) stores random vector for 1 layer. So the costs of DFA, FGT, and FGT (S) on GPU can be slightly larger than BP, which accords with the actual cost. Despite this, the overall costs are comparable.

On the other hand, if we consider neuromorphic computing with unidirectional synapses, BP (if possible) should maintain all backward synapses between successive layers, while FGT, FGT (S), and DFA keep direct feedback connections from the top layer, which can be smaller than BP since middle layers usually have much more neurons. As shown in Table 5, the memory costs of DFA and FGT (S) can be much smaller than BP.

Second, for operation numbers, BP requires a backward propagation across $N$ layers, DFA requires direct feedback to $N$ layers, FGT requires $N$ forward propagation of forward gradients for different layers (across $N, N-1, ..., 1$ layers) and direct feedback to $N$ layers, and FGT (S) requires a forward propagation across average $N/2$ layers and direct feedback to $N$ layers which can be more practical than FGT. Direct feedback to $N$ layers may need fewer operations than BP across $N$ layers as middle layers usually have more neurons (for example, propagation from 400 to 400 neurons requires much more operations than from 10 to 400 neurons). So with a rough estimation, DFA needs much fewer operations than BP, and FGT (S) needs fewer operations than BP, as shown in Table 5.

Additionally, we emphasize that our method mainly focuses on developing neuromorphic-friendly algorithms rather than only considering efficiency on GPU. Our ultimate goal is to consider efficient neuromorphic hardware, and as it is currently immature we develop methods with simulation on GPU. Our method is with more neuromorphic properties and is more plausible than BP. And compared with DFA-based methods, our method significantly improves performance to a similar level as BP. We can realize these targets with comparable costs, which could be acceptable.

## E.2 FIRING RATE STATISTICS

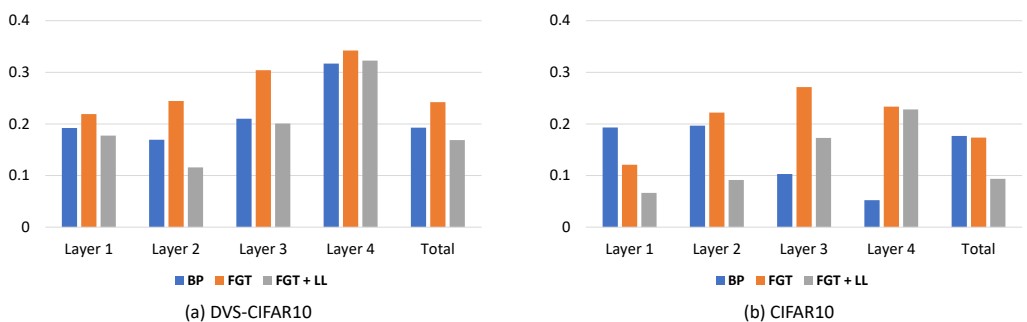

(a) DVS-CIFAR10

(b) CIFAR10

Figure 4: Firing rate statistics of models trained by different spatial credit assignment methods on DVS-CIFAR10 and CIFAR-10.

We compare the firing rate of models trained by BP, FGT, and FGT + LL in Fig. 4. On the neuromorphic dataset, the trend in different layers is similar for different methods, and the model trained

by FGT has more spikes than BP, while local learning can reduce the spikes. On the static dataset, the trend in different layers is different for BP and FGT, and the model trained by FGT has similar spikes as BP, while local learning can also significantly reduce the spikes by about half.

### E.3 RESULTS OF DEEPER NETWORKS

Table 7: Comparison results of a deeper 9-layer network (64C3-128C3-AP2-256C3-256C3-AP2-512C3-512C3-AP2-512C3-512C3-FC) on DVS-CIFAR10 and CIFAR-100 under the stochastic setting. IGL denotes intermediate global learning, where we add an additional readout layer at the middle of the network (4-th layer) that also provides feedback signals to previous layers.

| DVS-CIFAR10 | | | | CIFAR100 | | | |
|---|---|---|---|---|---|---|---|
| BP | DFA | FGT | FGT + LL | BP | DFA | FGT | FGT + IGL |
| 74.5 | 65.0 | 71.4 | 74.3 | 64.53 | 48.76 | 60.48 | 63.87 |

We supplement results with a deeper 9-layer network in Table 7. Our method is still valid. It is similar to the 5-layer setting that there is a large improvement over DFA ($> 10\%$ on CIFAR-100), and FGT with local learning (or intermediate global learning (IGL) with an additional readout layer at middle that also provides feedback to previous layers) achieves similar performance as BP.

The results show the potential for deeper networks. On the other hand, biological systems do not have that deep architectures, and previous works show that shallow ANNs with recurrence achieve higher functional fidelity of human brains and similarly high performance (Kubilius et al., 2019). Recent works also show that ANNs with about 10 layers can achieve SOTA results with special designs (Chen et al., 2023). So we think that the results provide effective verification of our method, and future work can consider advanced network architectures.

### E.4 MORE COMPARISON RESULTS

Table 8: Comparison results with the adjusted error feedback matrices method DKP.

| Method | MNIST | N-MNIST | DVS-Gesture | DVS-CIFAR10 | CIFAR-10 | CIFAR-100 |
|---|---|---|---|---|---|---|
| DFA | 97.83 | 97.68 | 82.87 | 60.20 | 79.92 | 50.09 |
| DKP | 97.80 | 97.26 | 44.10 | 36.80 | 82.86 | 53.78 |
| FGT | 98.12 | 97.98 | 95.60 | 73.30 | 85.98 | 61.19 |

We supplement comparison results with the adjusted error feedback matrices method DKP (Webster et al., 2020) in Table 8. DKP is based on the formulation of DFA and learn feedback weights similar to Kolen-Pollack learning, i.e., the gradient for the feedback weight is calculated based on the product of the middle layer's activation and the error from the top layer, whose thought is to keep the update direction of feedback and feedforward weights the same. DKP is designed for ANN, and we implement it for SNN with the adaptation of activations to pre-synaptic traces for feedback weight learning (similar to the update of feedforward weight). As shown in the results, DKP has around 3% performance improvement compared with DFA on CIFAR-10 and CIFAR-100, which is about the same as in its paper. However, we observe that DKP cannot work well for neuromorphic datasets. And both DKP and DFA have significant performance gaps compared with FGT.

### E.5 FINETUNING RESNET-34 ON IMAGENET UNDER NOISE

To further study the effectiveness of our method under large-scale and complex settings, we supplement an experiment on finetuning a pretrained ResNet-34 on ImageNet under potential hardware noise. This task is on the ground that there can be hardware mismatch for deploying trained SNN models (Yang et al., 2022; Cramer et al., 2022), and we may expect finetuning models directly on hardware to better deal with the problem. Our method paves paths for on-chip learning and may be combined with other works aiming at GPU training for SOTA performance in such scenario. We simulate a kind of potential hardware noise by considering the stochastic neuron setting as described

in Section 3.1, which may correspond to some hardware noise such as thermal noise. We consider finetuning a pretrained NF-ResNet-34 model released by Xiao et al. (2022), which is trained under the deterministic setting (i.e., no noise) with the original test accuracy as 65.15%, under the noise setting with different levels of noises. All methods are trained for only 1 epoch with the Adam optimizer and learning rate 2e-6, and the results are below (where the noise parameter corresponds to the hyperparameter for the sigmoid function of the logistic noise, and the larger the parameter the larger the noise scale).

Table 9: Comparison results of finetuning ResNet-34 on ImageNet under noise by different methods.

| Noise parameter | Direct test | DFA 1 epoch | FGT (S) 1 epoch | *BP 1 epoch* |
|:---:|:---:|:---:|:---:|:---:|
| 1/6 | 49.17 | 44.42 | **58.56** | *61.40* |
| 1/8 | 58.51 | 58.34 | **62.14** | *63.05* |

As shown in Table 9, our proposed FGT (S) can effectively finetune such large-scale models, while previous biologically plausible methods with random feedback (DFA) fail. BP is not biologically plausible and its results are only for reference. FGT (S) has lower performance than BP under the condition of 1 epoch because it may have a slower convergence rate than BP (as studied in Section 5). Our proof-of-concept results mainly verify the effectiveness of our method for such scenarios and large-scale models. Future work can consider more SOTA models, e.g., (Kim et al., 2022; Li et al., 2023).

Besides this setting, we may also expect potential on-chip training of pretrained models for new data or tasks considering real applications, as pretrain-finetune is the popular scenario considering recent advances in deep learning. These can be interesting future work.

## F  MORE DISCUSSIONS

### F.1  COMPARISON TO WORKS WITH SOTA PERFORMANCE

Our work is a different line from most recent works with SOTA performance (Kim et al., 2022; Li et al., 2023; Zhou et al., 2023), and they can be orthogonal and combined with each other. Specifically, recent works with SOTA performance focus on training high-performance SNNs on common hardware (e.g., GPU) with training methods such as BPTT and other improvements of models, and the trained models are expected to be directly deployed on neuromorphic hardware. Their hardware efficiency is about trained models. Our work, on the other hand, aims at developing algorithms that adhere to neuromorphic properties and are expected to pave paths for on-chip training of SNNs. Our hardware efficiency is about the training algorithm.

These two lines of work are compatible with each other. First, most recent SOTA works focus on the improvement of models and are still based on BP for training. Our work considers a different error propagation method and is orthogonal to them, which can be possibly combined in the future. With biologically plausible training methods that pave paths for potential on-chip training, we may also expect energy-efficient training directly on hardware, because GPUs are not designed for event-driven and in-memory computation, and training SNNs on GPUs is costing.

Second, considering real applications, we can also first pretrain a high-performance SNN on GPU as these works, and after deploying it on hardware, we may expect on-chip training of the models to deal with problems such as hardware mismatch or tuning the model for new data or tasks so that the model can adaptively evolve for real applications. One such scenario is verified in Appendix E.5.

Our experiments also mainly make fair comparisons with BP under the same settings and show the effectiveness of our method with better neuromorphic properties, without claiming superiority over SOTA models.

### F.2  NEUROMORPHIC COMPUTING PROPERTIES

Currently, the development of neuromorphic hardware is immature and is also developing considering software-hardware co-design (Schuman et al., 2022), so most algorithms are simulated on

common computational devices while considering properties of neuromorphic computing, without experiments on real neuromorphic chips. The compatibility is measured by fitness to neuromorphic properties (or biological plausibility) because neuromorphic hardware will be designed following them for efficient event-driven and in-memory computation. An important feature of neuromorphic computing is unidirectional synaptic connections for event-driven and in-memory computation to avoid frequent memory transport which accounts for the majority of energy costs in real hardware (the computation architecture is expected to be different from the commonly used hardware with von Neumann architecture such as CPU or GPU). This means that each synapse between two neurons only allows unidirectional communications, and its weight value is stored locally. This is the reason why BP can hardly be suitable for (future) neuromorphic hardware, because BP will require two different forward and backward connections between successive layers with exactly the same weight, which will need frequent memory exchange (and also two stages, etc.) that is not compatible to the basic thought of ideal neuromorphic computing, and can have larger costs even if it is supported. Our proposed FGT, on the other hand, avoids BP's problems and is thus more biologically plausible and compatible with neuromorphic hardware. We do not restrict the method to specific hardware, but follow the basic properties that will guide the development of hardware and algorithms.

### F.3  DISCUSSION OF LIMITATIONS

This work is built on online training methods for SNNs, so similar to these methods, it may also limit the usage of some techniques such as batch normalization along the temporal dimension. We follow Xiao et al. (2022) to adopt the scaled weight standardization for convolutional networks, while more techniques can be explored to better suit the properties of SNNs and scale up models. Additionally, this work mainly considers feedforward networks, and future work can consider more biologically plausible recurrent architectures with multiple layers.

