# OpenReview forum: "Forward Gradient Training of Spiking Neural Networks"
_ICLR.cc/2024/Conference — Submitted to ICLR 2024_

### Official Review · Reviewer_D141 · 2023-10-31

**Soundness:** 3 good
**Presentation:** 3 good
**Contribution:** 3 good
**Rating:** 5
**Confidence:** 4

**Summary:**

This article presents a novel approach for training Spiking Neural Networks (SNN) known as Forward Gradient Training (FGT). FGT introduces a spatial error signal allocation scheme, providing a solution to distribute errors across the network. By solely relying on forward propagation, FGT enables direct supervision signal transmission across layers, thus avoiding the layer-by-layer error propagation inherent in backpropagation. This methodology proves to be more conducive to online learning on chip architectures compared to traditional methods.

**Strengths:**

1. The writing in this article is commendable, exhibiting clear and fluid expression. The derivation of mathematical formulas is coherent, enhancing its readability significantly.
2. This article breaks free from the conventional framework of Backpropagation Through Time (BPTT) and boldly explores a new training approach. It introduces a more biologically plausible spatial error propagation method called Forward Gradient with Momentum Feedback Connections.

**Weaknesses:**

1. Compared to BPTT, FGT incurs larger training costs and is relatively slower on GPUs.
2. The article claims that this method takes a step forward towards online learning on chips, but it does not delve into the implementation of FGT on a chip or the associated storage and computational costs.

**Questions:**

1. The current article only validates the effectiveness of FGT on shallow and simpler network structures, showing comparable results to BPTT. However, its performance on deeper and more complex networks remains unexplored. The limited performance on smaller networks suggests that if this method fails to scale well to larger networks, the significance of this work may be limited. For example, the authors could discuss why the proposed framework provides a more feasible implementation of synchronous circuits. Also, there are existing methods that estimate the FLOPS and energy cost of a given NN based on its memory and operation. The authors could provide such comparisons as well. In addition, it is also necessary to provide the comparison with other  training methods on CIFAR-100 as well as ImageNet.

---

> ### Author Response · Authors · 2023-11-19
> **Response to Reviewer D141 (Part 1/2)**
>
> Thank you for your valuable comments. We respond to your comments and questions as follows. The citations are the same as the paper unless specified.
>
> 1. About training costs on GPU.
>
> First, our training cost comparison is with BP based on online training methods (Xiao et al., 2022) rather than BPTT. As described in the paper, our work is built upon online methods and the comparisons mainly focus on different spatial credit assignment methods given the online methods. For BPTT, the training memory cost is proportional to time steps, while online training methods have constant memory costs and is smaller than BPTT (e.g., for the considered 6 time steps, there can be $2-3\times$ reduction). Our method also has smaller memory costs than BPTT.
>
> Second, we would like to emphasize that our method mainly focuses on developing neuromorphic-friendly algorithms rather than only considering efficiency on GPU. Our ultimate goal is to consider efficient neuromorphic hardware, and as it is currently immature, we develop methods with simulation on GPU. Our method has more neuromorphic properties and is more plausible than BP. And compared with random-feedback-based methods (DFA), our method significantly improves performance to a similar level as BP. We can realize these targets with comparable costs, which could be acceptable.
>
> In Appendix E.1, we also discussed training costs conditions on potential neuromorphic computing hardware in the text, and our FGT (S) is better than BP (if possible) considering neuromorphic computing. On GPU, the training costs of FGT (or FGT(S)) is (slightly) larger than BP because GPU does not follow neuromorphic architectures and we do not optimize low-level codes of our method as BP in the established PyTorch libraries (the results are only brief comparisons). It can also be further improved, e.g., with parallelization of computation for forward gradients.
>
> Meanwhile, our developed variant FGT (S) is more practical in real applications and also has about the same costs and speed as BP on GPUs, as shown in Table 6 in Appendix E, which can also scale to large-scale networks as shown in the following response to the third point.
>
> 2. About discussion of implementation on a chip or the associated storage and computational costs.
>
> Currently, the development of neuromorphic hardware is immature and is also developing considering software-hardware co-design [1], so most algorithms are simulated on common computational devices while considering properties of neuromorphic computing, without experiments on real neuromorphic chips. The compatibility for implementation is measured by fitness to neuromorphic properties (or biological plausibility) because neuromorphic hardware will be designed following them for efficient event-driven and in-memory computation. FGT tackles BP’s problems of weight transport and separate phases for error signal propagation as it only requires unidirectional forward propagation across layers, conforming to properties of neuromorphic computing. We do not restrict the method to specific hardware, but follow the basic properties that will guide the development of hardware and algorithms. In Section 4.4, we also discussed some details on how the signal propagation of forward gradient can be made biologically plausible, proposing the variants of FGT (e.g., FGT (S), FGT (Q1, S)) which are also verified in experiments.
>
> We have discussed storage and computational costs on potential neuromorphic computing hardware in Appendix E.1, and we further supplement a comparison table for clearer presentation in this revision. We quote the discussions and the table here: “*for memory costs, … On the other hand, if we consider neuromorphic computing with unidirectional synapses, BP (if possible) should maintain all backward synapses between successive layers, while FGT, FGT (S), and DFA keep direct feedback connections from the top layer, which can be much smaller than BP since middle layers usually have much more neurons. As shown in Table 5, the memory costs of DFA and FGT (S) can be much smaller than BP*”, “*for operation numbers, BP requires a backward propagation across $N$ layers, DFA requires direct feedback to $N$ layers, FGT requires $N$ forward propagation of forward gradients for different layers (across $N, N−1, ..., 1$ layers) and direct feedback to $N$ layers, and FGT (S) requires a forward propagation across average $N/2$ layers and direct feedback to $N$ layers which is more practical than FGT. Direct feedback to $N$ layers may need fewer operations than BP across $N$ layers as middle layers usually have more neurons (for example, propagation from 400 to 400 neurons requires much more operations than from 10 to 400 neurons). So with a rough estimation, DFA needs much fewer operations than BP, and FGT (S) needs fewer operations than BP, as shown in Table 5*”.
>
> (continued below)

---

> > ### Author Response · Authors · 2023-11-19
> > **Response to Reviewer D141 (Part 1/2)**
> >
> > (continued from the preceding paragraph)
> >
> > | Method | Memory | Operations |
> > | :----: | :----: | :----: |
> > | BP (if possible) | $O((N-1)n^2+mn)$ | $O((N-1)n^2+mn)$ |
> > | DFA | $O(Nmn)$ | $O(Nmn)$ |
> > | FGT (S) | $O(Nmn)$ | $O(\frac{1}{2}(N-1)n^2+mn+Nmn)$ |
> >
> > The above table (Table 5 in Appendix E) is an estimation of training costs on potential neuromorphic hardware. For illustration, we consider neural networks with $N$ hidden layers with $n$ neurons for each layer and $m$ neurons for the output layer, where $m \ll n$. The costs mainly focus on additional synaptic costs besides the normal forward procedure. Our method FGT (S) is expected to be better than BP considering costs, and achieving much better performance than DFA.
> >
> > 3. About deeper and more complex networks.
> >
> > First, in Appendix E.3 (Table 7), we provided results for a deeper 9-layer convolutional network and our method is still effective and has a large improvement over DFA, with comparable performance as BP. The results show the potential for deeper networks. At the same time, biological systems do not have that deep architecture, and previous works show that shallow ANNs with recurrence achieve higher functional fidelity of human brains and similarly high performance (Kubilius et al., 2019). Future work can consider advanced network architectures.
> >
> > Second, to further study large-scale and complex settings, we supplement an experiment on finetuning a pretrained **ResNet-34 on ImageNet** under potential hardware noise. This task is on the ground that there can be hardware mismatch for deploying trained SNN models (Yang et al., 2022; [2]), and we may expect finetuning models directly on hardware to better deal with the problem. Our method paves paths for on-chip learning and may be combined with other works aiming at GPU training for SOTA performance in such scenario. We simulate a kind of potential hardware noise by considering the stochastic neuron setting as described in Section 3.1, which may correspond to some hardware noise such as thermal noise. We consider finetuning a pretrained ResNet-34 model released by Xiao et al. (2022), which is trained under the deterministic setting (i.e., no noise) with the original test accuracy as 65.15%, under the noise setting with different levels of noises. All methods are trained for only 1 epoch and the results are below (where the noise parameter corresponds to the hyperparameter for the sigmoid function of the logistic noise, the larger the parameter the larger the noise scale).
> >
> > | Noise parameter | Direct test | DFA 1 epoch | FGT (S) 1 epoch | *BP 1 epoch* |
> > | :----: | :----: | :----: | :----: | :----: |
> > | 1/6 | 49.17 | 44.42 | **58.56** | *61.40* |
> > | 1/8 | 58.51 | 58.34 | **62.14** | *63.05* |
> >
> > As shown in the results, our proposed FGT (S) can effectively finetune such large-scale model, while previous biologically plausible methods with random feedback (DFA) totally fail. BP is not biologically plausible and its results are only for reference. FGT (S) has lower performance than BP under the condition of 1 epoch because it may have a slower convergence rate than BP (as studied in Section 5). Our proof-of-concept results mainly verify the effectiveness of our method for such scenarios and large-scale models. Future work can consider more SOTA models.
> >
> > Besides this setting, we may also expect potential on-chip training of pretrained models for new data or tasks considering real applications, as pretrain-finetune is the popular scenario considering recent advances in deep learning. These can be interesting future work.
> >
> > 4. About why the proposed method is more feasible.
> >
> > Our proposed method tackles BP’s problems of weight transport and separate phases for error signal propagation as it only requires unidirectional forward propagation across layers, conforming to properties of neuromorphic computing.
> >
> > 5. About estimation of energy costs of a given NN.
> >
> > In Appendix E.2 (Figure 4), we have provided the firing rate statistics of models trained by different methods, which is usually used for energy estimation since the energy is proportional to spike counts. As shown in the results, on DVS-CIFAR10, the model trained by FGT has more spikes than BP, while with local learning, the spike number is reduced and is smaller than BP. On CIFAR10, the model trained by FGT has similar spikes as BP, while with local learning, the spike number is largely reduced by about half, implying fewer energy costs.
> >
> > 6. About comparison with other training methods on CIFAR-100 and ImageNet.
> >
> > In Section 5 (Table 3 and Table 4), we have already included comparison results on CIFAR-100.
> >
> > In the above response to the third point, we supplemented results on finetuning ResNet-34 on ImageNet under noises.
> >
> > [1] Schuman et al. Opportunities for neuromorphic computing algorithms and applications. Nature Computational Science, 2022.
> >
> > [2] Cramer et al. Surrogate gradients for analog neuromorphic computing. PNAS, 2022.

---

> > > ### Comment · Reviewer_D141 · 2023-11-20
> > > **Response to the rebuttal**
> > >
> > > Thanks for the explanation and for providing further results.  It seems that the efficacy of the method for large networks remains a concern. Regarding the feasibility of neuromorphic chips, bio-plausibility is not a main concern. The key issue is whether the algorithm is robust to the deviation in arrival time, which is typically not ensured in asynchronous circuits. It would be helpful if the authors could provide a discussion from this perspective with a detailed investigation of the executing steps of the proposed method.

---

> > > > ### Author Response · Authors · 2023-11-20
> > > >
> > > > Thank you for your further question.
> > > >
> > > > 1. Considering asynchronous circuits, our method, similar to DFA, can be better than BP since we do not need exact forward-backward synchronization or update blocking. For BP, it requires sequential layer-by-layer backward propagation after forward procedure to obtain gradients for each layer, and the update of weights should be blocked before backpropagation through it. Our method, on the other hand, simultaneously calculate and propagate forward gradients during forward propagation and do not require an additional synchronization stage for layer-by-layer backward propagation. The possible asynchronization for forward propagation will not introduce additional influence on our gradient calculation result (the asynchronization may influence the forward results of SNNs, but since our forward gradient is simultaneous with forward propagation, our gradient is also calculated under this condition and should be consistent). At the same time, our error signals are directly propagated from the top layer to all middle layers, similar to DFA, and can be parallel and asynchronous. Our method also does not impose restriction on the update of weights – theoretically, once the error signal is arrived no matter the time, the weights can be updated based on it and the local eligibility traces. While whether different learning speed (update numbers) for different layers will influence the optimization convergence is not thoroughly studied in existing works, it may possibly be solved by adaptive learning rates considering update frequency. So theoretically our method can be robust to asynchronization of both forward propagation and error propagation, which can be better than BP considering asynchronous circuits, and achieving much better performance than DFA.
> > > >
> > > > 2. About large networks. Our proof-of-concept results mainly verify the effectiveness of our method for large models like ResNet-34, while previous biologically plausible method totally fails. Usually, biologically more plausible algorithms do not reach BP efficiency considering convergence rate (e.g., the one epoch setting we consider). But as our brain also do not solely rely on supervised learning, it can be important future work to study better optimization objectives (e.g., local learning, self-supervision, etc.) that may be better combined with our method.

---

> > > > > ### Author Response · Authors · 2023-11-22
> > > > >
> > > > > Dear Reviewer D141,
> > > > >
> > > > > Thank you again for your efforts in reviewing our paper and providing valuable feedback. As the author-reviewer discussion phase is due soon, we would like to respectfully inquire if our further responses have adequately addressed your concerns or if there are any further questions that you would like us to address. In the further response, we supplemented the discussion on the advantage of our training method considering asynchronous circuits. We believe investigating new training algorithms with better neuromorphic properties and promising performance is important for SNNs. Could you please kindly reevaluate our paper? Thank you very much.

---

### Official Review · Reviewer_NYct · 2023-10-31

**Soundness:** 3 good
**Presentation:** 3 good
**Contribution:** 2 fair
**Rating:** 5
**Confidence:** 4

**Summary:**

The paper presents a FGT method for training SNNs that requires unidirectional FW prop and direct feedback from top layer.

**Strengths:**

The paper talks about bio-plausibility and hardware compatibility.

There are some theoretical guarantees that the paper talks about with performance better than random feedback.

The authors have shown their method works on simple datasets.

**Weaknesses:**

While the paper presents promising results, there may be an overstatement of the method's superiority without thorough comparison to a wide range of existing methods. There are substantial works [1-2] and many more that target hardware efficiency that have shown SOTA performance.
[1] Li, Yuhang, et al. "SEENN: Towards Temporal Spiking Early-Exit Neural Networks." arXiv preprint arXiv:2304.01230 (2023).
[2] Kim, Youngeun, et al. "Exploring lottery ticket hypothesis in spiking neural networks." European Conference on Computer Vision. Cham: Springer Nature Switzerland, 2022.
I am not sure if the authors have a quantitative way of suggesting their method's superiority to others in terms of accuracy or efficiency.

Finally, can the authors comment if their work has any limitations in terms of scalability or applicability to different data?

I am willing to change my rating post-rebuttal if the reviewers can give me a better justification about their results as compared to SOTA.

**Questions:**

See weaknesses above

---

> ### Author Response · Authors · 2023-11-19
> **Response to Reviewer NYct (Part 1/2)**
>
> Thank you for your valuable comments. We respond to your comments and questions as follows. The citations are the same as the paper unless specified.
>
> 1. About discussion and comparison to works with SOTA performance.
>
> Our work is a different line from these works, and they can be orthogonal and combined with each other. Specifically, recent works with SOTA performance focus on training high-performance SNNs on common hardware (e.g., GPU) with training methods such as BPTT and other improvements of models, and the trained models are expected to be directly deployed on neuromorphic hardware. Their hardware efficiency is about trained models. Our work, on the other hand, aims at developing algorithms that adhere to neuromorphic properties and are expected to pave paths for on-chip training of SNNs. Our hardware efficiency is about the training algorithm.
>
> These two lines of work are compatible with each other. First, most recent SOTA works focus on the improvement of models and are still based on BP for training. Our work considers a different error propagation method and is orthogonal to them, which can be possibly combined in the future. With biologically plausible training methods that pave paths for potential on-chip training, we may also expect energy-efficient training directly on hardware, because GPUs are not designed for event-driven and in-memory computation, and training SNNs on GPUs is costing.
>
> Second, considering real applications, we can also first pretrain a high-performance SNN on GPU as these works, and after deploying it on hardware, we may expect on-chip training of the models to deal with problems such as hardware mismatch or tuning the model for new data or tasks so that the model can adaptively evolve for real applications.
>
> To demonstrate the effectiveness for such scenario, we supplement an experiment on finetuning a pretrained ResNet-34 on ImageNet under potential hardware noise. This task is on the ground that there can be hardware mismatch for deploying trained SNN models (Yang et al., 2022; [1]), and we may expect finetuning models directly on hardware to better deal with the problem. We simulate a kind of potential hardware noise by considering the stochastic neuron setting as described in Section 3.1, which may correspond to some hardware noise such as thermal noise. We consider finetuning a pretrained ResNet-34 model released by Xiao et al. (2022), which is trained under the deterministic setting (i.e., no noise) with the original test accuracy as 65.15%, under the noise setting with different levels of noises. All methods are trained for only 1 epoch and the results are below (where the noise parameter corresponds to the hyperparameter for the sigmoid function of the logistic noise, the larger the parameter the larger the noise scale).
>
> | Noise parameter | Direct test | DFA 1 epoch | FGT (S) 1 epoch | *BP 1 epoch* |
> | :----: | :----: | :----: | :----: | :----: |
> | 1/6 | 49.17 | 44.42 | **58.56** | *61.40* |
> | 1/8 | 58.51 | 58.34 | **62.14** | *63.05* |
>
> As shown in the results, our proposed FGT (S) can effectively finetune such large-scale model, while previous biologically plausible methods with random feedback (DFA) totally fail. BP is not biologically plausible and its results are only for reference. FGT (S) has lower performance than BP under the condition of 1 epoch because it may have a slower convergence rate than BP (as studied in Section 5). Our proof-of-concept results mainly verify the effectiveness of our method for such scenarios and large-scale models. Future work can consider more SOTA models.
>
> We do not overstate our contribution as we do not aim at outperforming models with SOTA performance, but propose a new biologically more plausible training algorithm and show that it can achieve similarly promising results as BP which is adopted by training those models. Our experiments also mainly make fair comparisons with BP under the same settings and show the effectiveness of our method with better neuromorphic properties, without overstating superiority over SOTA models.
>
> We have added the above discussion and experiments, and cited your provided valuable references in the revised paper.
>
> 2. About the method’s superiority.
>
> Our method is superior to BP mainly because it tackles BP’s problems of weight transport and separate phases for error signal propagation as it only requires unidirectional forward propagation across layers, conforming to properties of neuromorphic computing. At the same time, our method can achieve comparable performance as BP, largely outperforming previous biologically plausible methods with random feedback. In short, the advantage lies in promising performance with better properties, paving paths to biologically plausible on-chip training of SNNs.

---

> > ### Author Response · Authors · 2023-11-19
> > **Response to Reviewer NYct (Part 2/2)**
> >
> > 3. About limitations in terms of scalability or applicability to different data.
> >
> > Our method is an alternative to BP and has the same applicability to different data as BP without additional limitations. As for scalability, our experiments reveal that our method may have a slower convergence speed than BP for convolutional networks, so this may be a limitation under the large-scale train-from-scratch setting. Despite this, as we have verified in experiments in the above response to the first point, our method can efficiently finetune pretrained ResNet-34 on ImageNet under noises with only 1 epoch. As pretrain-finetune is the popular scenario considering recent advances in deep learning, our work can be applied to such scenario for scalability, and also serves as an orthogonal line to recent SOTA SNN works as discussed in the above response to the first point.
> >
> > [1] Cramer et al. Surrogate gradients for analog neuromorphic computing. PNAS, 2022.

---

> > > ### Author Response · Authors · 2023-11-22
> > >
> > > Dear Reviewer NYct,
> > >
> > > Thank you again for your efforts in reviewing our paper and providing valuable feedback. As the author-reviewer discussion phase is due soon, we would like to respectfully inquire if our responses have adequately addressed your concerns or if there are any further questions that you would like us to address. In the response, we clarified that our work is orthogonal to and can be combined with works with SOTA performance such as your valuable references, and supplemented the proof-of-concept experiment as well as discussions on several aspects. Could you please kindly reevaluate our paper? Thank you very much.

---

### Official Review · Reviewer_Azqo · 2023-10-31

**Soundness:** 3 good
**Presentation:** 3 good
**Contribution:** 2 fair
**Rating:** 5
**Confidence:** 3

**Summary:**

The authors proposed forward gradient training (FGT) for spiking neural networks to perform biologically plausible and hardware friendly supervised learning. The key objective is how to deliver error signals at the last layer without backpropagating through previous layers. The authors developed forward surrogate gradient to address the non-differentiable functions of SNN and empirically demonstrated promising accuracy compared to BP and DFA. Also, FGT combined with local learning (LL), which optimizes the local loss estimated from local readout layers, shows more improved accuracy. Experiments are performed in various benchmark datasets, such as N-MNIST, DVS-Gesture, CIFAR-10, and CIFAR-100.

**Strengths:**

- The motivation of the authors is reasonable given many prior works on supervised SNN are still relying on backpropagation. Also, the biological plausibility and hardware compatibility are reasonable motivation to some extent to explore other supervised learning methods.

- The differences between FGT, backpropagation (BP), and direct feedback alignment (DFA) are well described in the figure and mathematically formulated. The paper is generally well written and provides enough information to understand the proposed approach with detailed appendix.

- FGT and LL couples the global and local supervised learning in SNN empirically showing good results. Table 3 and Figure 3 comprehensively summarize the performance of FGT and LL in the various datasets and comparison methods.

**Weaknesses:**

- The main concern is the novelty of the proposed method. The key contribution will be the section 4.1, where the authors present forward surrogate gradient to make forward gradient applicable to SNN. However, forward gradient, auto differentiation, and local learning are still existing concepts in the prior works.

- Despite the repeated emphasis on hardware compatibility and importance of on-chip training, the relevant experimental results are not presented. Appendix E discusses the training costs, but FGT is still worse than BP.

- The discussion on why BP is biologically implausible is enough, but why FGT and LL are biologically plausible and what are not plausible are not discussed clearly.

**Questions:**

- The higher training cost of FGT and LL makes sense to some extent as CPU/GPU are not SNN-friendly architectures. However, SNN accelerators are still being developed, and it is difficult say there is a standard architecture for neuromorphic computing. For example, if there are dedicated hardware accelerators for BP-based supervised SNN, BP will be more hardware compatible than FGT.

- It is not easy to make connections to well-known biological or biologically-inspired learning rules such as Hebbian learning or spike-time dependent plasticity (STDP). If the layer-wise error propagation is the main difference with BP, the proposed method is loosely related to biological learning rules.

---

> ### Author Response · Authors · 2023-11-19
> **Response to Reviewer Azqo (Part 1/2)**
>
> Thank you for your valuable comments. We respond to your comments and questions as follows. The citations are the same as the paper.
>
> 1. About novelty.
>
> Actually, our key contribution is **Section 4.2** where we propose a novel method for forward gradients, i.e., momentum feedback connections, while Section 4.1 is only our minor contribution that generalize the concept of forward gradient to the surrogate gradient condition of SNNs. The important problem of forward gradients in previous works (Silver et al., 2022; Baydin et al., 2022; Ren et al., 2023) is that the vanilla forward gradient is a projection to a random direction, so it suffers from large variance and performs poorly, which is also verified in our experiments. To tackle the problem, in Section 4.2, we propose the novel forward gradient with momentum feedback connections, which can largely reduce the variance and significantly improve the performance, also with theoretical analysis. With the proposed method, our FGT can achieve a similar performance as BP, which is **not** achieved by all previous works on forward gradients. This is one of our key contributions. Additionally, another novel contribution is presented in **Section 4.4**, where we also propose variants of our method with more biologically plausible considerations (e.g., FGT (S), FGT (Q1, S)) and we conduct experiments to verify their effectiveness, which has not been considered by previous works. Our method is novel and different from existing forward gradient methods.
>
> 2. About results considering hardware compatibility and training costs.
>
> Currently, the development of neuromorphic hardware is immature, so most algorithms are simulated on common computational devices while considering properties of neuromorphic computing, without experiments on real neuromorphic chips. The compatibility is measured by fitness to neuromorphic properties (or biological plausibility) because neuromorphic hardware is designed following these properties for efficient event-driven and in-memory computation. Note that an important feature of neuromorphic computing is *unidirectional* synaptic connections for event-driven and *in-memory* computation to avoid frequent memory transport which accounts for the majority of energy costs in real hardware (the computation architecture is expected to be different from the commonly used hardware with von Neumann architecture such as CPU or GPU). This means that each synapse between two neurons only allows unidirectional communications, and its weight value is stored locally. This is the reason why BP can hardly be suitable for (future) neuromorphic hardware, because BP will require two different forward and backward connections between successive layers with exactly the same weight, which will need frequent memory exchange (and also two stages, etc.) that is not compatible to the basic thought of neuromorphic computing. Our proposed FGT (and variants), on the other hand, avoids BP’s problems and is thus more biologically plausible and compatible with neuromorphic hardware.
>
> As for the discussion on training costs in Appendix E, Table 6 is only a rough comparison on GPU and we have also discussed conditions on potential neuromorphic computing hardware in the text, where our FGT (S) is better than BP (if possible) considering neuromorphic computing. We also supplement a comparison table (Table 5 in the revised version, also shown below) for clearer presentation of this. On GPU, the training costs of FGT (or FGT(S)) is (slightly) larger than BP because GPU does not follow neuromorphic architectures and we do not optimize low-level codes of our method as BP in the established PyTorch libraries (the results are only brief comparisons). It can also be further improved, e.g., with parallelization of computation for forward gradients.
>
> Meanwhile, our developed variant FGT (S) is more practical in real applications and also has about the same training costs and time as BP on GPUs, as shown in Table 6 in Appendix E.1.
>
> (continued below)

---

> ### Author Response · Authors · 2023-11-19
> **Response to Reviewer Azqo (Part 2/2)**
>
> (continued from the preceding paragraph)
>
> Considering neuromorphic computing, FGT (S) can be better and we quote our discussion and comparison table in Appendix E.1 here:
> “*for memory costs, … On the other hand, if we consider neuromorphic computing with unidirectional synapses, BP (if possible) should maintain all backward synapses between successive layers, while FGT, FGT (S), and DFA keep direct feedback connections from the top layer, which can be much smaller than BP since middle layers usually have much more neurons. As shown in Table 5, the memory costs of DFA and FGT (S) can be much smaller than BP*”, “*for operation numbers, BP requires a backward propagation across $N$ layers, DFA requires direct feedback to $N$ layers, FGT requires $N$ forward propagation of forward gradients for different layers (across $N, N−1, ..., 1$ layers) and direct feedback to $N$ layers, and FGT (S) requires a forward propagation across average $N/2$ layers and direct feedback to $N$ layers which can be more practical than FGT. Direct feedback to $N$ layers may need fewer operations than BP across $N$ layers as middle layers usually have more neurons (for example, propagation from 400 to 400 neurons requires much more operations than from 10 to 400 neurons). So with a rough estimation, DFA needs much fewer operations than BP, and FGT (S) needs fewer operations than BP, as shown in Table 5*”.
>
> | Method | Memory | Operations |
> | :----: | :----: | :----: |
> | BP (if possible) | $O((N-1)n^2+mn)$ | $O((N-1)n^2+mn)$ |
> | DFA | $O(Nmn)$ | $O(Nmn)$ |
> | FGT (S) | $O(Nmn)$ | $O(\frac{1}{2}(N-1)n^2+mn+Nmn)$ |
>
> The above table (Table 5 in Appendix E.1) is an estimation of training costs on potential neuromorphic hardware. For illustration, we consider neural networks with $N$ hidden layers with $n$ neurons for each layer and $m$ neurons for the output layer, where $m \ll n$. The costs mainly focus on additional synaptic costs besides the normal forward procedure. Our method FGT (S) is expected to be better than BP considering costs, and achieving much better performance than DFA.
>
> Additionally, we also would like to emphasize that our method mainly focuses on developing neuromorphic-friendly algorithms rather than only considering efficiency on GPU. We realize our targets, i.e., more neuromorphic properties and a similar level of performance as BP, with comparable costs, which could be acceptable.
>
> 3. About why FGT and LL are plausible.
>
> FGT tackles BP’s problems of weight transport and separate phases for error signal propagation as FGT only requires unidirectional forward propagation across layers, conforming to properties of neuromorphic computing. This makes FGT more biologically plausible. LL is also more biologically plausible than BP because it is local learning without signal backpropagation across layers, avoiding the above problems of BP.
>
> Considering what is not plausible, the original FGT requires propagating real-valued forward gradients for all layers, which may be hard to realize. So in Section 4.4, we have further discussed how the signal propagation of forward gradient can be made plausible with bursts and random sampling of layers, proposing the variants of FGT (e.g., FGT (S), FGT (Q1, S)) which are also verified in experiments.
>
> 4. About whether BP can be more hardware compatible than FGT.
>
> While it is true that neuromorphic hardware is still being developed, BP is not a good choice because it is not compatible with the basic thought of neuromorphic computing and can have larger costs even if it is supported (e.g., frequent memory exchange, more synapses considering forward and backward between all successive layers, etc.), as discussed in the above response to the second point. Our FGT (and variants) is not specific to certain hardware, but follows the basic properties of neuromorphic computing that will guide the development of hardware, which may be more suitable. We believe that developing biologically plausible training methods is important for SNNs.
>
> 5. About connection to biologically-inspired learning rules.
>
> In Section 4.3, we have discussed the connection to three-factor Hebbian learning. Our method is based on previous online training methods of SNNs, which is connected to the three-factor rule, while our method further answers how the global modulator can be determined – in our method, it is directly propagated from the top layer, corresponding to a more plausible neuro-modulation than previous methods that should be backpropagated across layers.

---

> > ### Author Response · Authors · 2023-11-22
> >
> > Dear Reviewer Azqo,
> >
> > Thank you again for your efforts in reviewing our paper and providing valuable feedback. As the author-reviewer discussion phase is due soon, we would like to respectfully inquire if our responses have adequately addressed your concerns or if there are any further questions that you would like us to address. In the response, we clarified the novelty and our key contributions, and supplemented discussions on several aspects. Could you please kindly reevaluate our paper? Thank you very much.

---

### Author Response · Authors · 2023-11-19
**A summary of paper updates**

We thank all reviewers for their valuable feedback. We have uploaded the updated version of our paper based on the reviews. Revisions are marked as blue in the text. The updates are summarized as follows:

1. In response to Reviewer Azqo and D141, we reorganize the discussion of training costs in Appendix E.1, showing the difference between costs on potential neuromorphic hardware and GPU, where our proposed FGT (S) can be better than BP in the former condition which is our target. We also added more discussions on the properties of neuromorphic computing in Appendix F.

2. In response to Reviewer D141 and NYct, we supplement an experiment on finetuning a pretrained ResNet-34 on ImageNet under potential hardware noise in Appendix E.5, in order to show the scalability to deeper models and the scenario that our method can be combined with works aiming at training SNNs on GPUs with SOTA performance.

3. In response to Reviewer NYct, we added citations as well as discussions on recent works with SOTA performance in Appendix F, clarifying that our method can be orthogonal to and combined with them.

Detailed responses to every question are in each separate response to reviewers.

---

### Meta-Review · Area_Chair_dJDm · 2023-12-12

**Metareview:**

The paper discusses a new algorithm for training Spiking Neural Networks, called Forward Gradient Training, so that to be able to compute gradients to the last layer without having access to computations to earlier layers (local learning only). The main objective is to derive algorithms that are eventually friendly to (currently not-mature) neuromorphic, 'biologically plausible' architectures.

Reviewers see the paper warmly, however, they all raise points of criticism.
- Is the method novel enough?
- Is the method validated empirically well enough on large networks and datasets?
- Is the neuromorphic attributes well tested?

I find the answers in the right direction. Being able to scale to a ResNet-34 would certainly give a quite convincing answer. The fact that the training is one a single epoch, however, makes it hard to tell whether the experiment is fully done, as the training could still easily converge to a poor local minimum.

I thus suggest a more finalized empirical verification, as well as a better explained description of novelty compared to previous works. Then paper should be ready to be accepted.

**Justification For Why Not Higher Score:**

Experiments in the rebuttal are not completely ready yet.

**Justification For Why Not Lower Score:**

See above.

---

### Decision · Program_Chairs · 2024-01-16

Reject